# FUNDAMENTAL LIMITS ON THE ROBUSTNESS OF IMAGE CLASSIFIERS

**Zheng Dai & David K. Gifford**
Computer Science and Artificial Intelligence Laboratory
Massachusetts Institute of Technology
Cambridge, MA 02139, USA
`{zhengdai,gifford}@mit.edu`

## ABSTRACT

We prove that image classifiers are fundamentally sensitive to small perturbations in their inputs. Specifically, we show that given some image space of $n$-by-$n$ images, all but a tiny fraction of images in any image class induced over that space can be moved outside that class by adding some perturbation whose $p$-norm is $O(n^{1/\max(p,1)})$, as long as that image class takes up at most half of the image space. We then show that $O(n^{1/\max(p,1)})$ is asymptotically optimal. Finally, we show that an increase in the bit depth of the image space leads to a loss in robustness. We supplement our results with a discussion of their implications for vision systems.

## 1 INTRODUCTION

Image classification, the task of partitioning images into various classes, is a classical problem in computer science with countless practical applications. Progress on this problem has advanced with leaps and bounds since the advent of deep learning, with modern image classifiers attaining some incredible results(Beyer et al., 2020). However, it has been observed that image classes tend to be brittle - classifiers like to partition images in a way such that most images lie very close to images of different classes(Szegedy et al., 2013). Although usually studied in computer vision systems, such phenomena also appear to manifest in natural vision systems(Elsayed et al., 2018; Zhou & Firestone, 2019).

Given these observations, it is natural to ask the following question: is the brittleness of image classes a result of classifier construction, or does it arise from some fundamental property of image spaces? Previous work demonstrate that classifiers can be made more robust as a function of how they are constructed, and attempts to improve the robustness of existing computer vision systems through such means is an active area of research(Moosavi-Dezfooli et al., 2016; Madry et al., 2017; Ma et al., 2018; Tramer et al., 2020; Machado et al., 2021). However, there is also a fundamental limit to the robustness achievable by *any* classifier that arises as a consequence of the geometry of image spaces.

In this work we show that this fundamental limit of achievable robustness is surprisingly low. Roughly speaking, in most cases it suffices to change the contents of only a few columns of pixels in an image to change its class. Even smaller changes are sufficient when measured using other metrics, such as the Euclidean metric. Our results are a consequence of the geometry of image spaces, and so they apply regardless of the architecture of the classifier. This suggests that there is an inherent brittleness in the semantic content of images, and that robustness as an objective is only desirable with respect to distributions that are concentrated over small subsets of the image space.

### 1.1 OUR CONTRIBUTIONS AND RELATED WORK

The observation that image classes tend to be brittle was popularized by Szegedy et al. (2013), where it was observed that tiny perturbations suffice to change the image class of many images. This has since opened up a rich field of research on how the brittleness of image classes arise from specific classifier formulations or training distributions (Goodfellow et al., 2014; Gilmer et al., 2018; Tsipras

et al., 2018)[1]. While these advances offer insights into the deficiencies of our current methodologies, their analyses ultimately depend on some aspect of the architecture or training distribution, so do not rule out the possible existence of ideal classifiers that do not induce brittle image classes.

By contrast, our work provides a non-trivial upper bound to the robustness of *any* image class[2]. Specifically:

- We prove that most images in any image class consisting of at most half the images in an image space of $n$-by-$n$ images can be moved into a different class by adding a perturbation whose $p$-norm is $\mathcal{O}(n^{1/\max(p,1)})$. This is a vanishingly small quantity in relation to the average distances in the image space, which is $O(n^{2/\max(p,1)})$, and therefore provides a non-trivial upper bound for the robustness attainable by even an ideal classifier.

- We show that there exist image classes where most images cannot be moved into a different class with any perturbation whose $p$-norm is $o(n^{1/\max(p,1)})$ (note the small-$o$ notation[3]). Therefore, the bound we derive is asymptotically optimal in $n$, so proving stronger robustness bounds will *require* examining classifier-specific properties.

- We show that discretization through lowering the bit depth of the image space permits the existence of more robust image classes. This lends theoretical backing to the idea of using discretization as a method of defending against adversarial attacks (Panda et al., 2019).

- We demonstrate that brittle features in images can deliver semantic content. We argue that a deeper understanding of such features can lead to advances in aligning human and computer vision systems.

To our knowledge, there are two previous works that investigate upper bounds of robustness that arise from the geometry of image spaces. One is from Fawzi et al. (2018a), which provides an upper bound for the probability that an image drawn from a given distribution is far from images of a different class. They further perform numerical experimental analyses of their bounds. However, our analysis differs and improves on theirs in a few key aspects. Firstly, they only analyze the case where distance is measured using the 2-norm, while we provide bounds for $p$-norms for any $p$. Secondly, they do not account for the discrete nature of image spaces with finite bit depth, which allows for classifiers that are more robust than their bounds imply[4]. Finally, their bound is parametrized by a modulus of continuity which differs depending on the image distribution, potentially resulting in trivial bounds for certain distributions. Furthermore, this parameter cannot be computed exactly, so in application their bound is inexact. By contrast, we formulate our results independently of specific image distributions[5]. Our bounds can therefore be computed exactly and unconditionally, and we are able to show the asymptotic optimality of our result.

The other work is from Diochnos et al. (2018), which investigates partitions of bit vectors. Since bit vectors can be used to encode discrete inputs, their results can be viewed as results about classifiers over discrete inputs. They also view each bit vector as being equally weighted, so their results are not parametrized by data distributions and are unambiguous. They show that given a finite probability of misclassification, an arbitrarily high proportion of vectors can be turned into misclassified vectors through small numbers of bit flips proportional to the square root of the vector dimension. This result has been generalized in follow-up work (Mahloujifar et al., 2019), where it was shown that a small number of modifications proportional to the square root of the data dimension suffices to induce misclassification in the more general setting of Lévy families as well. However, these results are still dependent on the existence of a finite fraction of misclassified datapoints, and therefore do not preclude the existence of asymptotically infinitesimal image classes that are robust, something which

---

[1]Goodfellow et al. (2014) demonstrate how linearity can allow imperceptible changes across a large amount of dimensions to accumulate to something significant, Gilmer et al. (2018) derive a relation between classification error rate and the distance to the closest misclassification on a specific dataset of concentric spheres, and Tsipras et al. (2018) derive a fundamental tradeoff between accuracy and robustness.

[2]Consisting of at most half the images in the image space.

[3]$f(n) \in \mathcal{O}(g(n)) \iff \limsup_{n\to\infty} \frac{f(n)}{g(n)} \leq \infty$ and $f(n) \in o(g(n)) \iff \lim_{n\to\infty} \frac{f(n)}{g(n)} = 0$

[4]Other work, such as that of Diochnos et al. (2018), does analyze discrete input spaces, but does not investigate the relation between discretization and robustness.

[5]It is still possible adapt our result to account for image distribution using their techniques. See our discussion for additional details.

our analysis does preclude. Furthermore our bounds, due to our focus on image spaces, are much stronger than the ones they derive and are asymptotically optimal.

The observation that brittle features can deliver semantic content is an observation on the inadequacy of $p$-norms for quantifying visual similarity. While such inadequacies have been noted in prior work (Tramèr et al., 2020; Fawzi et al., 2018b), to our knowledge ours is the first that have been derived as a consequence of theoretical bounds. The theoretical foundation of our observations allows for quantification and has the potential to imply further consequences.

## 2 RESULTS

### 2.1 PRELIMINARIES

Images consist of pixels on a two dimensional grid, with each pixel consisting of a set of channels (for example R, G, and B) of varying intensity. We therefore define the *image space of $h$-channel images of height $n$ and aspect ratio $q$*, denoted $\mathcal{I}_{n,q,h,(\infty)}$, as the set of all real valued tensors with shape $(qn, n, h)$ with entries lying in the interval $[0, 1]$. We require that $qn$ be an integer[6]. The first two dimensions index the $x$ and $y$ coordinate of the pixel, while the third indexes the channel.

Only a finite subset of these images can be represented with finite bit strings. Therefore, we use $\mathcal{I}_{n,q,h,(b)}$ to denote the subset of $\mathcal{I}_{n,h,q,(\infty)}$ where each entry in the tensor is one of $2^b$ equally spaced values between 0 and 1 inclusive (in other words, each entry belongs to $[0, 1] \cap \{i/(2^b - 1) | i \in \mathbb{Z}\}$). We will refer to $b$ as the bit depth of the image space. Although technically image spaces of equal height, aspect ratio, and number of channels intersect, we will treat them as disjoint (in other words, if $x \in \mathcal{I}_{n,q,h,(b)}$, then $x \notin \mathcal{I}_{n',q',h',(b')}$ if $\mathcal{I}_{n,q,h,(b)} \neq \mathcal{I}_{n',q',h',(b')}$).

Images in the image space $\mathcal{I}_{n,q,h,(b)}$ contain $n^2qh$ entries. This quantity $n^2qh$ appears often in our results[7] and can be thought of as the data dimension (the number of dimensions required to specify the data), which we will often denote with $N$ for simplicity. The data dimension can be set to any integer value (for example by setting $n = 1$ and $q = 1$), so any data type that can be represented as a Cartesian product of some fixed number of unit intervals $[0, 1]$ can be viewed as an image as we have defined it. Consequently our theoretical results can be applied to a more general class of inputs than images, though we will continue to focus on the case of image classification in this work.

#### 2.1.1 CLASSIFIERS AND CLASSES

A classifier $\mathcal{C}$ is a function $\mathcal{I}_{n,q,h,(b)} \to \mathcal{Y}$, where $\mathcal{Y}$ is some finite set of labels. For each $y \in \mathcal{Y}$, we define the class of $y$ as the preimage of $y$, denoted as the set of images $\mathcal{C}^{-1}(y)$. We say that such a class is *induced* by $\mathcal{C}$. If a class takes up a large part of the image space, then it contains a lot of images that look like randomly sampled noise, since randomly sampling channel values from a uniform distribution yields a uniform distribution over the image space. Therefore, many images in these classes tend to be uninteresting, which motivates the following definition:

**Definition 1.** *A set $C \subseteq \mathcal{I}_{n,q,h,(b)}$ is an interesting image class if it is not empty, and if it contains no more than half of the total number of images in $\mathcal{I}_{n,q,h,(b)}$.*

Note that as long as every class of a classifier is populated, no more than one class can be uninteresting since image classes are disjoint. Therefore, if a classifier does induce an uninteresting class, we can think of it as *the* uninteresting class. Intuitively we can think of it as a junk class, since it contains images that look like randomly sampled noise.

Most of our results will pertain to interesting image classes. This is to eliminate pathological considerations such as when an image class covers the entire image space.

---

[6]We opt to use $qn$ rather than a separate value for image width to suggest that the height and width of an image should have similar magnitudes.

[7]This is because our results depend only on the size of the image tensor rather than its shape.

### 2.1.2 Perturbations and Robustness

In order to discuss perturbations, we define addition and subtraction over tensors that are of the same shape to be element-wise, and we define the $p$-norm of a tensor $A$, denoted $\|A\|_p$, to be the $p$th root of the sum of the absolute values of the entries of $A$ raised to the $p$th power. $p$ is assumed to be a non-negative integer, and for the special case of $p = 0$ we let $\|A\|_0$ be the number of non-zero entries in $A$. Note that when $p$ is 0 the 0-"norm" is not truly a norm since it does not obey homogeneity.

We can then define what it means for an image to be robust to perturbations:

**Definition 2.** *Let $C \subseteq \mathcal{I}_{n,q,h,(b)}$ be a class of images. We say an image $I \in C$ is robust to $L^p$-perturbations of size $d$ if for all $I' \in \mathcal{I}_{n,q,h,(b)}$, $\|I - I'\|_p \leq d$ implies $I' \in C$.*

We can then define what it means for a class to be robust to perturbations. Note that unless a class occupies the entire image space, it must contain some non-robust images, so the best we can hope for is to attain robustness for a large fraction of the images within a class. This is reflected in the following definition.

**Definition 3.** *Let $C \subseteq \mathcal{I}_{n,q,h,(b)}$ be a class of images. Then we say that $C$ is $r$-robust to $L^p$-perturbations of size $d$ if it is not empty, and the number of images $I \in C$ that are robust to $L^p$-perturbations of size $d$ is at least $r|C|$, where $|C|$ is the number of images in $C$.*

## 2.2 Universal upper bound on classifier robustness

We can now state a universal non-robustness result that applies to all classifiers over discrete image spaces $\mathcal{I}_{n,q,h,(b)}$.

**Theorem 1.** *For all real values $c > 0$, there exists no interesting class $C \subseteq \mathcal{I}_{n,q,h,(b)}$ (the image space of $h$-channel images of height $n$ and aspect ratio $q$ with bit depth $b$) that is $2e^{-2c^2}$-robust to $L^p$-perturbations of size $(2 + c\sqrt{N})^{1/\max(p,1)}$, where $N = n^2qh$ is the data dimension.*

*Proof sketch.* We can use the images in $\mathcal{I}_{n,q,h,(b)}$ to form a graph where images are the vertices, and images are connected if and only if they differ at exactly one channel. In other words, the image tensors must differ at precisely one entry. Figure 1a illustrates the construction of this graph. Note that graph distance between vertices coincides with the Hamming distance between the images represented by the vertices. Such graphs are known as Hamming graphs, and they have a vertex expansion (or isoperimetry) property (Harper, 1999) which implies that for any sufficiently small set, if we add all vertices that are within a graph distance of $\mathcal{O}(\sqrt{N})$ to that set, then the size of that set increases by at least some given factor (see Figure 1b for an example). This expansion property is contingent on the size of the vertex set being sufficiently small, which is why we require the "interesting class" property.

We can then show that an interesting class $C$ cannot be too robust in the following way: suppose for contradiction that it is. Then there must be some set $C' \subseteq C$ that is pretty large, and has the property that all vertices within some graph distance of $C'$ are in $C$. We can then use the vertex expansion property to show that adding these vertices to $C'$ gives a set larger than $C$, which contradicts the assumption that all vertices within some graph distance to $C'$ are in $C$. Plugging explicit values into this argument yields the statement of the theorem.

We can then generalize to $L^p$-perturbations for arbitrary $p$ since each coordinate varies by at most 1 unit. The full proof can be found in Appendix A.1. □

Intuitively, the above results state that to change the class of most "interesting" images, the number of pixels that need to be changed is roughly the number contained in a few columns of the image.

### 2.2.1 The universal non-robustness results are asymptotically optimal up to a constant factor

Up to a constant factor, the bounds in Theorem 1 are the best possible for a universal non-robustness result that applies to arbitrary predictors if we only consider the data dimension $N$ and hold the bit depth $b$ constant. In other words, there exists no bound on robustness that applies *universally to all classifiers* that grows much more slowly in $N$ than the ones given in Theorem 1. Therefore, if we wish

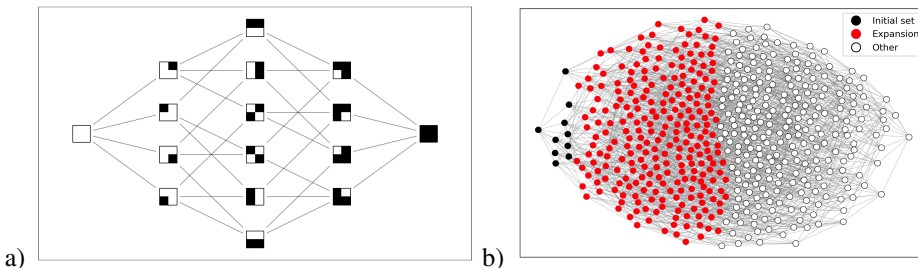

Figure 1: Interpreting image spaces as Hamming graphs

a) We show how we construct a Hamming graph using the elements of $\mathcal{I}_{2,1,1,(1)}$, the space of binary images on four pixels. By construction, graph distance coincides exactly with Hamming distance.
b) We demonstrate the expansion property of Hamming graphs on a Hamming graph constructed using $\mathcal{I}_{3,1,1,(1)}$ as the vertex set. If we pick some initial set of vertices (in black), then the set of vertices that are a graph distance of at most 3 (3 being the image height $n$ in this case) from that initial set (in black and red) is much larger than that initial set. The nature of "much larger" is expanded on in Appendix A.1.

to show that the classes induced by some classifier are not robust to, for instance, $L^0$-perturbations of size $\mathcal{O}(log(N))$, more specific properties of that classifier would need to be considered.

To prove this, consider the classifier defined by Algorithm 1[8].

---

**Algorithm 1:** Robust Classifier

---

**Input :** An image $I \in \mathcal{I}_{n,q,h,(b)}$
**Result:** A label belonging to $\{0, 1\}$
$S \leftarrow 0$;
**for** $x \leftarrow 1$ **to** $qn$ **do**
    **for** $y \leftarrow 1$ **to** $n$ **do**
        **for** $a \leftarrow 1$ **to** $h$ **do**
            $S \leftarrow S + I_{x,y,a}$;

**if** $S < n^2qh/2$ **then**
    **return** 0;
**else**
    **return** 1;

---

**Theorem 2.** *The classifier described by Algorithm 1 induces an interesting class $C \subseteq \mathcal{I}_{n,q,h,(b)}$ (the image space of $h$-channel images of height $n$ and aspect ratio $q$ with bit depth $b$) such that for all $c > 0$:*

*1. $C$ is $(1 - 4c)$-robust to $L^p$-perturbations of size $c\sqrt{N} - 2$ for all $p \leq 1$.*

*2. $C$ is $(1 - 4c)$-robust to $L^p$-perturbations of size $\frac{(c\sqrt{N}-2)^{1/p}}{(2^b-1)^{(p-1)/p}}$ for all $p \geq 2$.*

*Where $N = n^2qh$ is the data dimension.*

*Proof sketch.* Given an image $I$, let $S(I)$ be the sum of all its channel values subtracted by $N/2$ (where $N = n^2qh$ is the data dimension). Then $I$ being robust to $L^1$-perturbations of size $x$ is approximately equivalent to $S(I) \notin [-x, x]$. By the central limit theorem, the fraction of images $I$ such that $S(I) \notin [-c\sqrt{N}, c\sqrt{N}]$ is some monotonic function of $c$ independent of $N$ if $N$ is sufficiently large, which is our desired result. Appendix A.2 provides a more careful analysis of this that does not rely on limiting behaviour and extends the result to all $p$-norms. $\square$

---

[8]In Einstein notation, the algorithm returns $I_{x,y,a}\mathbf{1}^{x,y,a} \geq n^2qh/2$ on an image tensor $I$, where $\mathbf{1}$ is shaped like $I$ and has 1 at each entry. We spell out the algorithm in pseudocode for clarity.

We remark that the $c$ in Theorem 2 should be set to less than $1/4$ in order to yield a non-trivial statement.

### 2.2.2 CLASSIFIER ROBUSTNESS TO $L^p$-PERTURBATIONS DECREASES WITH INCREASING BIT DEPTH FOR $p \geq 2$

In this section we investigate the role played by the bit depth $b$. Theorem 2 has a dependency on $b$ when considering $L^p$-perturbations for $p \geq 2$, so the statement about attainable robustness becomes increasingly vacuous as the bit depth increases. Somewhat surprisingly, this is not an artifact of suboptimal analysis: it is really the case that the fundamental limits of robustness drops as a function of the bit depth of the image space.

**Theorem 3.** *For all real values $c > 0$ and $p \geq 2$, no interesting class $C \in \mathcal{I}_{n,q,h,(b)}$ (the image space of $h$-channel images of height $n$ and aspect ratio $q$ with bit depth $b$) is $2e^{-c^2/2}$-robust to $L^p$-perturbations of size $\left(c + 2\frac{\sqrt{N}}{2^b}\right)^{2/p}$, where $N = n^2qh$ is the data dimension.*

*Proof sketch.* We will focus on the 2-norm. Extension to higher $p$-norms is straightforward and is given as part of the full proof found in Appendix A.3. The main idea of the proof rests on the fact that if we extend the classifier to the continuous image space with something like a nearest neighbour approach, the measure of the images that are robust to perturbations of a constant size is small (the statement and proof may be found in Appendix A.4). Therefore, if we randomly jump from an image in the discrete image space to an image in the continuous image space, with high probability we will be within a constant distance of an image in a different class. The size of this random jump can be controlled with a factor that shrinks with increasing bit depth. Summing up the budget required for this jump, the perturbation required on the continuous image space, and the jump back to the discrete image space yields the desired bound. $\square$

We remark that this suggests that the bounds in Theorem 1 pertaining to $L^p$-perturbations for $p \geq 2$ can be improved to reflect its dependency on the bit depth $b$. However, whether the component that shrinks with $b$ scales with $N^{1/2p}$ rather than $N^{1/p}$ remains an open problem.

## 3 DISCUSSION

### 3.1 SUMMARY OF ROBUSTNESS LIMITS AND THEIR RELATION TO AVERAGE IMAGE DISTANCES

We summarize the bounds we derived in the previous section in Table 1, where the bounds are reparametrized in terms of the robustness. An asymptotic bound is also provided with respect to the data dimension $N$ and bit depth $b$. A plot of the relation between perturbation sizes and robustness can be found in Appendix A.6.

With respect to $N$ and $b$, the bounds derived for the 0-norm and 1-norm are asymptotically optimal, while the bounds for the other $p$-norms are asymptotically optimal with respect to $N$. Finding the optimal bound with respect to both $b$ and $N$ for $p$-norms for $p \geq 2$ remains an open problem, although our current analysis suffices to show that $b$ does fundamentally influence how robust an interesting class can be.

We note that the bounds we derived are vanishingly small when compared to typical distances between random elements of the image space. If a pair of images $I, I' \in \mathcal{I}_{n,q,h,(b)}$ are sampled independently and uniformly, we have:

$$\mathbb{E}[\|I - I'\|_p] \geq k_{b,p} N^{1/\max(1,p)} \tag{1}$$

Where $k_{b,p}$ is some constant parametrized by $b$ and $p$, and $N = n^2qh$ is the data dimension. See Appendix A.5 for additional details. While our analysis there does not necessarily hold for images drawn non uniformly, we demonstrate in Appendix A.5.1 that typical distances on natural distributions tend to be similar in size.

When this observation is combined with the bounds in Table 1, we can see that when $N$ is sufficiently large, for 99% (or some arbitrarily high percentage) of images $I''$ within an interesting class $C$:

Table 1: Fundamental bounds for robustness attainable by any interesting image class in $\mathcal{I}_{n,q,h,(b)}$. $N = n^2 qh$ is the data dimension. Rather than leaving the robustness and bound parametrized by a separate constant $c$, the bounds have been reparametrized in terms of the robustness $r$. Figures plotting the relation between the bounds and $r$ can be found in Appendix A.6. The bounds are also given in big-$\Theta$ notation, where $r$ is held constant for simplicity. The upper bound should be understood as "no interesting class is $r$-robust to perturbations of these sizes" and the lower bound should be interpreted as "there exists an interesting class that is $r$-robust to perturbations of these sizes".

| PERTURBATION | UPPER BOUND | LOWER BOUND |
|---|---|---|
| $L^0$-PERTURBATION $L^1$-PERTURBATION | $2 + \sqrt{\frac{1}{2} ln(\frac{2}{r})} \sqrt{N}$ | $-2 + (\frac{1-r}{4}) \sqrt{N}$ |
| $L^p$-PERTURBATION, $p \geq 2$ | $\min\left( \left(2 + \sqrt{\frac{1}{2} ln(\frac{2}{r})} \sqrt{N}\right)^{1/p}, \left(\sqrt{2 ln(\frac{2}{r})} + \frac{1}{2^{b-1}} \sqrt{N}\right)^{2/p} \right)$ | $\dfrac{\left( -2 + (\frac{1-r}{4}) \sqrt{N} \right)^{1/p}}{(2^b - 1)^{(p-1)/p}}$ |

| PERTURBATION | UPPER BOUND | LOWER BOUND |
|---|---|---|
| $L^0$-PERTURBATION $L^1$-PERTURBATION | $\Theta(\sqrt{N})$ | $\Theta(\sqrt{N})$ |
| $L^p$-PERTURBATION, $p \geq 2$ | $\Theta(\min(N^{1/2p}, (\frac{\sqrt{N}}{2^b} + 1)^{2/p})$ | $\Theta(\frac{N^{1/2p}}{2^{b((p-1)/p)}})$ |

$$\frac{\min_{X \in \mathcal{I}_{n,q,h,(b)}, X \notin C} \|I'' - X\|_p}{\mathbb{E}[\|I - I'\|_p]} \leq c_{b,p} N^{-\frac{1}{2\max(p,1)}} \tag{2}$$

Where $c_{b,p}$ is some constant parametrized by $b$ and $p$.

The right hand side approaches 0 as $N$ grows without bound, so compared to typical distances one finds in an image space, the distance of an image to an image outside of its class is vanishingly small in any $p$-norm it is measured in.

## 3.2 IMPLICATIONS OF ROBUSTNESS LIMITS FOR COMPUTER VISION SYSTEMS

According to our results, a large fraction of images in any interesting image class are can have their classes modified by a small perturbation. We note that we only consider the case of image *classifiers* that partition images into a finite number of discrete classes, so our results do not apply directly to vision models that output class probabilities or some more abstract representation. However, these outputs must ultimately be converted into decisions when the model is deployed, at which point our bounds do apply.

Taken at face value, our results appear to pose a barrier for the construction of reliable computer vision systems. For illustration, suppose we implement a system that selects from a finite pool of actions to take depending on the output of some image classifier. Then for most images, a tiny perturbation can make the given image trigger undesired behaviour, ostensibly making the classifier unreliable.

One way of circumventing this barrier is to consider reliability conditioned on a prespecified image distribution. Our bounds do not immediately preclude the existence of small fractions of images within interesting image classes that are robust to large perturbations, and it is possible that those are precisely the set of images that are commonly encountered in deployment. Therefore, our bounds do not directly prevent the construction of classifiers that are robust with respect to some given image distribution, which is an ongoing field of research(Madry et al., 2017).

We remark that our bounds can be converted into bounds that take an image distribution into account through tricks similar to ones used by Fawzi et al. (2018a). We can define a map $f : \mathcal{I}_{n,q,h,(b)} \to \mathcal{I}_{n,q,h,(b)}$ such that $f(X)$ approximates the desired image distribution if $X$ is distributed uniformly over $\mathcal{I}_{n,q,h,(b)}$ (we can think of the preimage as some latent space). If we then define a monotonically increasing $\omega : \mathbb{R} \to \mathbb{R}$ such that $\|f(X) - f(X')\|_p \leq \omega(\|X - X'\|_p)$ for all pairs of images, then all our bounds stating that no interesting class is $r$-robust to $L^p$-perturbations of size $d$ can be converted into statements about how the probability of encountering an image in an interesting class that is robust to $L^p$-perturbations of size $\omega(d)$ is less than $r$. Clearly the value of such a bound necessarily depends on $f$ and $\omega$, and it is easy to construct examples where these bounds must be vacuous - for example, we could make $f$ map all images of one class to the entirely black image and all images of the other class to the entirely white image. However, this is an unrealistic toy example, and experimental analysis carried out by Fawzi et al. (2018a) suggest that analyzing plausible approximations of $f$ that produce useful distributions can produce informative bounds.

However, we also note that formulating robustness in a distribution specific way does not address all reliability concerns: for example, an ostensibly benign image modified imperceptibly to trigger dangerous behaviour can be a reliability concern regardless of whether said image is drawn from a prespecified distribution. Furthermore, identifying the correct distribution of such images is a non-trivial task, and distribution gaps can lead to significant reductions in performance(Recht et al., 2019).

Another way of circumventing the barrier to constructing reliable computer vision systems imposed by our bounds is to note that a perturbation with a small $p$-norm is not necessarily imperceptible. In these cases, the classifier *ought* to adjust its output with respect to such perturbations. Robustness should then be defined with respect to a perception aligned metric, as opposed to with some $p$-norm. The barrier to constructing reliable computer vision systems then becomes the alignment problem for human and computer vision systems.

### 3.3 BRITTLE FEATURES CAN IMPART SEMANTICALLY SALIENT INFORMATION

We show in this section that the semantic contents of most images are contained within brittle features that can be erased with a small perturbation. First, we note that our bounds apply universally to any image classifier, so they must apply to an *ideal* classifier that is able to faithfully mimic human classifications. Although such a classifier has yet to be achieved in the computer vision space, human based classifiers can be built quite easily: simply place people in front monitors and ask them to apply labels to images[9]. Memoization could be applied to prevent the same image from being classified multiple times with conflicting labels. This system, on top of producing human classifications, acts like a classifier which partitions the set of all images into disjoint classes (as described in Section 2.1.1), therefore our bounds must be satisfied[10].

At this point, we have proven the existence of a classifier that reproduces human classifications. To simplify the discussion, we will consider classifiers with only two classes: given a set of words[11], we will consider the class of images that such a label can be applied to and its complement. Specifically, the words should be chosen in a way such that random noise should not receive a label with high probability. Roughly speaking, the idea we attempt to capture here is that the first class is the class of "meaningful" images, and the latter is the class of "meaningless" images.

Since images drawn from random noise should not receive a label with high probability, the class of meaningless images is far larger than the class of meaningful images. Therefore, the class of meaningful images is an interesting class, and all our bounds apply. This has the surprising consequence that the semantic content of most images can be "erased" with a small perturbation by turning them into meaningless images. Put differently, the semantic contents of most meaningful images are contained in brittle features that can be erased with a small perturbation, which is what we set out to show.

---

[9]Such constructions have been commercially implemented for purposes like content moderation.

[10]It may be the case that a pair of counterfactual trajectories of classifications would yield contradictory labels on certain images. However, since only one such trajectory can occur factually, memoization suffices to make such a classifier observationally indistinguishable from the kind of classifiers discussed in Section 2.1.1.

[11]For example, words like "truck" or "parachute".

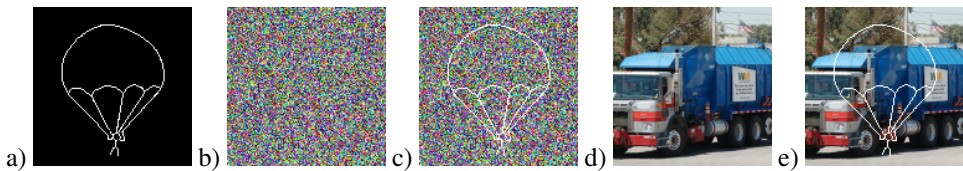

Figure 2: Small perturbations can be semantically salient

A small perturbation (a) when applied to a uniformly randomly drawn image (b) can add meaning to it (c). Conversely the meaning present in (c) can be removed with a small perturbation to attain (b). Such a perturbation can also add information to a natural image[a] (d, e) - although the information present in the natural image largely remains after applying the perturbation, new information was still added by the perturbation.

---

[a]Sourced from from (Howard)

What do these features look like? A possible example of brittle features that convey semantic information is line drawings (see Figure 2). Line drawings can be erased with a perturbation of size $\mathcal{O}(n)$ (proportional to the height of the image) when measured using the 1-norm or 0-norm, which is congruent with the bounds we derived for those metrics. Line drawings are also semantically salient. However, the size of a line drawing is generally larger than $\mathcal{O}(1)$ when measured with a $p$-norm with $p \geq 2$, so the saliency of line drawings do not necessarily account for our bit depth dependent bounds. Phenomena like optical illusions and pareidolia may offer some hints as to how saliency can be present in even more brittle features. Some past work highlight other possible ideas, such as the object of interest only taking up a small portion of the picture (Tramèr et al., 2020), or salient patterns being drawn with low opacity (Fawzi et al., 2018b). However, a full understanding remains elusive.

Understanding the ways in which brittle features can carry semantic meaning is not merely of academic curiosity. We have shown that the robustness of computer vision systems have fundamental limits. However, a computer vision system that is aligned to the human visual system *ought* to obey these limits, since a human based classifier must do so as well. Over the past decade we have learned that standard machine learning methodology does not automatically produce vision systems that are aligned to the human visual system with respect to small perturbations (Szegedy et al., 2013), and methodologies that seek to produce such vision systems still contain misalignments (Tramer et al., 2020). A deeper understanding of how semantic content is conveyed in brittle features may inform the development of future methodologies (for example we may wish to explicitly train computer vision systems on such features), which is becoming increasingly necessary as computer vision systems become increasingly deployed in safety and security critical applications, where the trustworthiness of the system is essential (Pereira & Thomas, 2020; Ma et al., 2018).

## 4 CONCLUSION

We have derived universal non-robustness bounds that apply to any arbitrary image classifier. We have further demonstrated that up to a constant factor, these are the best bounds attainable with respect to the dimensions of the image. These bounds provide fundamental limits to the robustness achievable by computer vision systems, and reveal that most images in any interesting class, even those induced by ideal classifiers, can have their class changed with a perturbation that is asymptotically infinitesimal when compared to the average distance between images. We then discuss the barriers to constructing safe and secure computer vision systems imposed by our results and how these barriers may be circumvented. Finally, we show the abundance of brittle features that convey semantic information, and propose that an improved understanding of these features may yield progress on the problem of aligning human and computer vision systems. We discuss line drawings as an attractive candidate for brittle features that are semantically salient. However, they are not sufficiently brittle to account for all our bounds, so a full understanding of these features remains the subject of future work.

## 5 REPRODUCIBILITY STATEMENT

Complete proofs for our claims can be found in the appendix.

## 6 ACKNOWLEDGEMENT

This work was supported in part by Schmidt Futures.

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

## A  APPENDIX

### A.1  PROOF OF THEOREM 1

#### A.1.1  PROPERTIES OF BINOMIAL COEFFICIENTS

We will work with binomial coefficients extensively. To simplify some of our statements, we will extend the definition of a binomial coefficient to work with any $n > 0$ and arbitrary integer $k$:

$$\binom{n}{k} = \begin{cases} \dfrac{n!}{k!(n-k)!} & \text{if } 0 \leq k \leq n \\ 0 & \text{otherwise} \end{cases} \tag{3}$$

Binomial coefficients can be bound in the following way:

**Lemma 1.** $\binom{n}{k} < \dfrac{2^n}{\sqrt{n}}$ *when* $n \geq 1$.

*Proof.* We first note that $n!$ is bounded by the following for all $n \geq 1$ (Robbins, 1955):

$$\sqrt{n}\frac{n^n}{e^n} < \frac{n!}{\sqrt{2\pi}} < \sqrt{n}\frac{n^n}{e^n}e^{1/(12n)} \tag{4}$$

Applying the appropriate inequalities for the numerator and denominator yields the following for when $n$ is even:

$$\binom{n}{k} \leq \binom{n}{n/2} = \frac{n!}{((n/2)!)^2} < 2\frac{2^n}{\sqrt{n}}\frac{e^{1/(12n)}}{\sqrt{2\pi}} \tag{5}$$

When $n$ is odd, we have:

$$\binom{n}{k} \leq \binom{n}{\lfloor n/2 \rfloor} \tag{6}$$

$$= \frac{1}{2} \binom{n+1}{(n+1)/2} \tag{7}$$

$$< 2 \frac{2^n}{\sqrt{n+1}} \frac{e^{1/(12(n+1))}}{\sqrt{2\pi}} \tag{8}$$

$$< 2 \frac{2^n}{\sqrt{n}} \frac{e^{1/(12n)}}{\sqrt{2\pi}} \tag{9}$$

Where the third comparison is an application of Equation 5.

If $n \geq 1$, we have $\frac{e^{1/(12n)}}{\sqrt{2\pi}} < 0.5$, which proves the claim. $\qquad\square$

It will also be useful to define the following cumulative sums (which are also the tails of binomial distributions):

$$U_{n,p}(k) = \begin{cases} \sum_{i=0}^{k} \binom{n}{i} p^i (1-p)^{n-i} & \text{if } k \geq 0 \\ 0 & \text{otherwise} \end{cases} \tag{10}$$

We can show that the ratio of these cumulative sums are monotonic increasing:

**Lemma 2.** *Let $p \in (0,1)$. Then $\frac{U_{n,p}(x-k)}{U_{n,p}(x)}$ is monotonic increasing in $x$, where $0 \leq x \leq n$ and $k$ is any positive integer.*

*Proof.* First, we note that the ratio $\binom{n}{x-k}/\binom{n}{x}$ is monotonic increasing in $x$ when $x \geq 0$. This holds by definition if $x - k < 0$. Otherwise, we have the following:

$$\frac{\binom{n}{x-k}/\binom{n}{x}}{\binom{n}{x-k+1}/\binom{n}{x+1}} = \frac{(n-x)}{(n-x+k)} * \frac{(x-k+1)}{(x+1)} \leq 1 \tag{11}$$

We then claim the following holds for all $x$ where $0 \leq x \leq n-1$:

$$\frac{U_{n,p}(x-k)}{U_{n,p}(x)} \leq \frac{U_{n,p}(x-k+1)}{U_{n,p}(x+1)} \leq \frac{\binom{n}{x-k+1}(1-p)^k}{\binom{n}{x+1}p^k} \tag{12}$$

The above holds with equality when $x - k + 1 < 0$. If $x - k + 1 = 0$, the above also holds: the leftmost ratio is 0. For the other two ratios, if we multiply the rightmost ratio by $(1-p)^{n-k}$ above we can see that the numerators are equal while the denominator of the rightmost ratio is smaller. We can prove the other cases by induction on $x$:

$$\frac{U_{n,p}(x-k)}{U_{n,p}(x)} \leq \frac{\binom{n}{x-k}(1-p)^k}{\binom{n}{x}p^k} \tag{13}$$

$$\leq \frac{\binom{n}{x-k+1}(1-p)^k}{\binom{n}{x+1}p^k} \tag{14}$$

$$= \frac{\binom{n}{x-k+1}p^{x-k+1}(1-p)^{n-x+k-1}}{\binom{n}{x+1}p^{x+1}(1-p)^{n-x-1}} \tag{15}$$

Where the first inequality follows by induction, and the second inequality follows because $\binom{n}{x-k}/\binom{n}{x}$ is monotonic increasing in $x$.

For any positive numbers $a$, $c$ and strictly positive numbers $b$, $d$, where $\frac{a}{b} \leq \frac{c}{d}$, we have $\frac{a}{b} \leq \frac{a+c}{b+d} \leq \frac{c}{d}$ because:

$$\frac{d}{d\lambda}\left(\frac{a+\lambda c}{b+\lambda d}\right) = \frac{bc - ad}{(b+\lambda d)^2} \geq 0 \tag{16}$$

Therefore, we have:

$$\frac{U_{n,p}(x-k)}{U_{n,p}(x)}$$

$$\leq \frac{U_n(x-k) + \binom{n}{x-k+1}p^{x-k+1}(1-p)^{n-x+k-1}}{U_{n,p}(x) + \binom{n}{x+1}p^{x+1}(1-p)^{n-x-1}} \tag{17}$$

$$= \frac{U_{n,p}(x-k+1)}{U_{n,p}(x+1)} \tag{18}$$

$$\leq \frac{\binom{n}{x-k+1}(1-p)^k}{\binom{n}{x+1}p^k} \tag{19}$$

As claimed. Carrying on the induction up to $x = n - 1$ yields the statement. $\square$

### A.1.2  BOUNDING THE INTERIOR OF A SET OVER A HAMMING GRAPH

We will prove our main results by an application of isoperimetry bounds over a Hamming graph. Let $W$ be a set of $w$ symbols. Then we define the $n$ dimensional Hamming graph over $w$ letters, denoted $\mathcal{H}(n, w)$, as the graph with a vertex set $W^n$ and an edge set containing all edges between vertices that differ at precisely one coordinate. For example, $\mathcal{H}(n, 2)$ is isomorphic to the Boolean hypercube. We will use $V(\mathcal{H}(n, w))$ to denote the vertex set of the Hamming graph.

Let $S \subseteq \mathcal{H}(n, w)$. We define the expansion of $S$, denoted $\text{EXP}(S)$, as the set of vertices that are either in $S$ or have a neighbour in $S$. Since $\text{EXP}(.)$ inputs and outputs sets of vertices, we can iterate it. We will use $\text{EXP}^k(.)$ to denote $k$ applications of $\text{EXP}(.)$.

We now adapt a a result from (Harper, 1999) (Theorem 3 in the paper). Additional details on how it has been adapted can be found in Appendix B.1.

**Lemma 3** (Isoperimetric Theorem on Hamming graphs). *Let $S \subsetneq \mathcal{H}(n, w)$. Then:*

$$\frac{|\text{EXP}^k(S)|}{|V(\mathcal{H}(n, w))|} \geq \min\{U_{n,p}(r+k)$$

$$|U_{n,p}(r) = \frac{|S|}{|V(\mathcal{H}(n, w))|},$$

$$p \in (0, 1), r \in [0, n-k)\} \tag{20}$$

To work with this we first obtain bounds for the expression on the right hand side of Lemma 3.

**Lemma 4.** *Let $p$ be any value in $(0, 1)$. Let $n > r \geq k$ such that $U_{n,p}(r) \leq \frac{1}{2}$. Then $\frac{U_{n,p}(r-k)}{U_{n,p}(r)} \leq 2e^{-2(\max(k-1,0))^2/n}$.*

*Proof.* Let $X$ be a binomially distributed random variable with $n$ trials and probability of success $p$. Let $m$ be the median of $X$. We have $m \leq np + 1$ because the median and mean differ by at most 1 (Kaas & Buhrman, 1980).

$U_{n,p}(m-k)$ can be interpreted as $\Pr(X \leq m-k)$, We can then apply Hoeffding's inequality (Hoeffding, 1994):

$$\Pr(X \le m - k) \le \Pr(X \le np + 1 - k) \tag{21}$$

$$\le e^{-2(\max(k-1,0))^2/n} \tag{22}$$

Since $m$ is the median of $X$, we also have $U_{n,p}(m) \ge \frac{1}{2}$. Combining this with the above equation gives:

$$\frac{U_{n,p}(m-k)}{U_{n,p}(m)} \le 2e^{-2(\max(k-1,0))^2/n} \tag{23}$$

Since $\frac{U_{n,p}(x-k)}{U_{n,p}(x)}$ is monotonically increasing in $x$ via Lemma 2, this also implies that the above relation holds for all $r \le m$. This completes the proof. □

We can then plug this into Lemma 3 to obtain a non-robustness result on Hamming graphs, which we will then apply to image spaces.

**Theorem 4.** *Let $S \subsetneq V(\mathcal{H}(n,w))$ such that $|S| \le |V(\mathcal{H}(n,w))|/2$, and $c > 0$ be any number. Let $S' \subseteq S$ be the set of vertices for which no path with $c\sqrt{n} + 2$ edges or less leads to a vertex not in $S$. Then $\frac{|S'|}{|S|} < 2e^{-2c^2}$.*

*Proof.* Suppose for contradiction that $|S'| \ge 2e^{-2c^2}|S|$. Since for any vertex in $S'$ no path with $c\sqrt{n} + 2$ edges or less leads to a vertex outside of $S$, we have $\text{EXP}^{c\sqrt{n}+2}(S') \subseteq S$. Then:

$$|\text{EXP}^{c\sqrt{n}+2}(S')| \ge |V(\mathcal{H}(n,w))| \min\{U_{n,p}(r + c\sqrt{n} + 2)$$

$$|U_{n,p}(r) = \frac{|S'|}{|V(\mathcal{H}(n,w))|},$$

$$p \in (0,1), r \in [0, n - c\sqrt{n} - 2]\} \tag{24}$$

$$\ge \frac{1}{2} e^{2(\max(c\sqrt{n}+1,0))^2/n}|S'| \tag{25}$$

$$> \frac{1}{2} e^{2c^2}|S'| \tag{26}$$

The first relation follows from Lemma 3 and the second follows from Lemma 4. Lemma 4 applies since $\text{EXP}^{c\sqrt{n}+2}(S') \subseteq S$, so $|\text{EXP}^{c\sqrt{n}+2}(S')| \le |S| \le \frac{1}{2}$.

But then $|\text{EXP}^{c\sqrt{n}+2}(S')| > \frac{1}{2}e^{2c^2}|S'| \ge |S|$, which implies that $\text{EXP}^{c\sqrt{n}+2}(S') \nsubseteq S$. This is a contradiction, so we obtain our desired statement. □

### A.1.3 PROVING THEOREM 1

All that remains is to massage Theorem 4 into the form of Theorem 1. Let $C \subseteq \mathcal{I}_{n,q,h,(b)}$ be any interesting class.

**Lemma 5.** *$C$ is not $2e^{-2c^2}$-robust to $L^0$-perturbations of size $c\sqrt{qh} * n + 2$.*

*Proof.* This is a straightforward corollary of Theorem 4 since we can construct a hamming graph out of $\mathcal{I}_{n,q,h,(b)}$ as shown in Figure 1 in the main text.

In detail: let $\mathcal{M} : V(\mathcal{H}(n^2qh, 2^b)) \to \mathcal{I}_{n,q,h,(b)}$ be the following bijection: first let $Q$ be a set of $2^n$ equally spaced values between 0 and 1, where the largest value is 0 and the smallest is 1. Then the elements of $V(\mathcal{H}(n^2qh, 2^b))$ can be viewed as $Q^{n^2qh}$. We then map elements from $Q^{n^2qh}$ to $\mathcal{I}_{n,q,h,(b)}$ such that the inverse operation is a flattening of the image tensor. Note that such a mapping preserves graph distance on $V(\mathcal{H}(n^2qh, 2^b))$ as Hamming distance on $\mathcal{I}_{n,q,h,(b)}$.

Let $C' \subseteq C$ be the set of images that are robust to $L^0$-perturbations of size $c\sqrt{qh} * n + 2$. Let $S = \mathcal{M}^{-1}(C)$ and $S' = \mathcal{M}^{-1}(C')$. $S'$ is then the set of vertices for which no path with $c\sqrt{qh} * n + 2$ edges or less leads to a vertex outside of $S$.

$C$ is an interesting class and $\mathcal{M}(.)$ preserves cardinality due to it being a bijection. Therefore $|C'| \leq |V(\mathcal{H}(n^2qh, 2^b))|/2$, so by Theorem 4 we have $|S'|/|S| < 2e^{-2c^2}$. Again, since $\mathcal{M}(.)$ preserves cardinality, this implies that $|C'|/|C| < 2e^{-2c^2}$, which means that $C$ is not $2e^{-2c^2}$-robust to $L^0$-perturbations of size $c\sqrt{qh} * n + 2$. $\qquad\square$

We remark that if the domain of $\mathcal{M}(.)$ is changed to $\mathcal{H}(qn^2, h2^b)$, the above argument also shows that $C$ is not $2e^{-2c^2}$-robust to $c\sqrt{q}n + 2$ pixel changes.

It is straightforward to generalize this to $p$-norms with larger $p$.

**Lemma 6.** *$C$ is not $2e^{-2c^2}$-robust to $L^p$-perturbations of size $(c\sqrt{qh} * n + 2)^{1/p}$.*

*Proof.* Let $S_1$ be the set of images that are $r$-robust to $L^0$-perturbations of size $d$, and let $S_2$ be the set of images that are $r$-robust to $L^p$-perturbations of size $d^{1/p}$.

Suppose $I \notin S_1$. Then there exists some image $I'$ in a different class from $I$ such that $\|I - I'\|_0 \leq d$. Therefore, for all $p > 0$, we have:

$$d \geq \|I - I'\|_0 \tag{27}$$

$$= \sum_{x,y,c} \lceil |I_{x,y,c} - I'_{x,y,c}| \rceil \tag{28}$$

$$\geq \sum_{x,y,c} |I_{x,y,c} - I'_{x,y,c}|^p \tag{29}$$

$$= (\|I - I'\|_p)^p \tag{30}$$

Where the second and third relation follows from the fact that channel values are contained in $[0, 1]$. Therefore, $I \notin S_2$ either since $\|I - I'\|_p \leq d^{1/p}$. Taking the contraposition yields $S_2 \subseteq S_1$.

Setting $d = c\sqrt{qh} * n + 2$ and applying Lemma 5 gives the desired result. $\qquad\square$

### A.2 PROOF OF THEOREM 2

#### A.2.1 ANTI-CONCENTRATION INEQUALITIES

We first prove an anti-concentration lemma concerning the binomial distribution.

**Lemma 7.** *Let $X$ be a random variable following the binomial distribution with $n$ trials and a probability of success of 0.5. Let $Y$ be a discrete random variable independent of $X$ whose distribution is symmetric about the origin. Then for any $t$ where $t < \mathbb{E}[X]$ and $t - \lfloor t \rfloor = 1/2$, we have:*

$$Pr(X + Y \leq t) \geq Pr(X < t) \tag{31}$$

*Proof.* We have the following:

$$Pr(X + Y \leq t) = Pr(X + Y \leq t, X < t) \tag{32}$$
$$+ Pr(X + Y \leq t, X > t)$$
$$Pr(X < t) = Pr(X + Y \leq t, X < t) \tag{33}$$
$$+ Pr(X + Y > t, X < t)$$

Therefore it suffices to show that $\Pr(X + Y \le t, X > t) \ge \Pr(X + Y > t, X < t)$. We have for any $r \ge 0$:

$$\Pr(X + Y \le t, X = t + r) = \Pr(Y \le -r)\Pr(X = t + r) \tag{34}$$
$$\ge \Pr(Y > r)\Pr(X = t + r) \tag{35}$$
$$\ge \Pr(Y > r)\Pr(X = t - r) \tag{36}$$
$$= \Pr(X + Y > t, X = t - r) \tag{37}$$

Where Equation 34 follows from the independence of $X$ and $Y$, Equation 35 follows from the symmetry of the distribution of $Y$, and Equation 36 follows from our assumption that $t < \mathbb{E}[X]$ and $t - \lfloor t \rfloor = 1/2$.

Summing over all positive $r$ for which $\Pr(X = t + r) \ge 0$ yields the desired result. $\square$

**Lemma 8.** *Let $X_1, X_2, ..., X_n$ be independently and identically distributed random variables such that each $X_i$ is uniformly distributed on $2k$ evenly spaced real numbers $a = r_1 < r_2 < ... < r_{2k} = b$. Then for $t > 0$, we have:*

$$Pr(\sum_{i=1}^{n} X_i \le (\sum_{i=1}^{n} \mathbb{E}[X_i]) - t + (b - a)) > \frac{1}{2} - \frac{2t}{\sqrt{n}(b - a)} \tag{38}$$

*Proof.* Let $Y_1, Y_2, ..., Y_n$ be independently and identically distributed Bernoulli random variables with $p = 0.5$. Let $Z_1, Z_2, ..., Z_n$ be a set of independently and identically distributed random variables uniformly distributed between the integers between $1$ and $k$ inclusive. If the $Y$s and $Z$s are independent of each other as well, we have:

$$\sum_{i=1}^{n}(X_i - \mathbb{E}[X_i]) = \frac{b - a}{2k - 1} \sum_{i=1}^{n}(kY_i + Z_i - \mathbb{E}[kY_i + Z_i]) \tag{39}$$

$$= k\frac{b - a}{2k - 1}((\sum_{i=1}^{n} Y_i) + (\sum_{i=1}^{n} \frac{Z_i - \mathbb{E}[Z_i]}{k})$$

$$- (\sum_{i=1}^{n} \mathbb{E}[Y_i])) \tag{40}$$

Let $\sum_{i=1}^{n} Y_i = B$, $\sum_{i=1}^{n} \frac{Z_i - \mathbb{E}[Z_i]}{k} = D$, and $k\frac{b-a}{2k-1} = c$. Then for any $t > 0$, we have:

$$\Pr(\sum_{i=1}^{n}(X_i - \mathbb{E}[X_i]) \leq -t) = \Pr(B + D \leq -\frac{t}{c} + \mathbb{E}[B]) \tag{41}$$

$$\geq \Pr(B + D$$
$$\leq -\frac{t}{c} + \mathbb{E}[B] - u) \tag{42}$$

$$\geq \Pr(B < -\frac{t}{c} + \mathbb{E}[B] - 1) \tag{43}$$

$$\geq \Pr(B - \mathbb{E}[B]$$
$$< -\frac{2t}{b-a} - 1) \tag{44}$$

$$\geq \frac{1}{2} - \Pr(B - \mathbb{E}[B]$$
$$\in [-\frac{2t}{b-a} - 1, 0]) \tag{45}$$

$$\geq \frac{1}{2} - \binom{n}{\lfloor n/2 \rfloor} 2^{-n}$$
$$(\frac{2t}{b-a} + 2) \tag{46}$$

Where $1 \geq u \geq 0$ is chosen such that $-\frac{t}{c} + \mathbb{E}[B] - u$ is the average of two adjacent integers. Equation 43 is then an application of Lemma 7 since $B$ is binomially distributed with $p = 0.5$ and $D$ has a distribution that is symmetric about the origin, and Equation 46 follows from the fact that no more than $x + 1$ values are supported on an interval of length $x$, and no supported value has probability greater than $\binom{n}{\lfloor n/2 \rfloor} 2^{-n}$.

Observing that $\binom{n}{\lfloor n/2 \rfloor} 2^{-n} < \frac{1}{\sqrt{n}}$ due to Lemma 1 and substituting $t$ with $t - (b - a)$ yields the desired result. $\qquad \square$

### A.2.2 PROVING THEOREM 2

Let $A : \mathcal{I}_{n,q,h,(b)} \to \{0, 1\}$ be described by Algorithm 1. In other words, it is the classifier that inputs an image, sums all of its channels, and outputs 0 if the sum is less than $n^2 qh/2$ and 1 otherwise. Let $Z$ be the class of images that $A$ outputs 0 on. Note that $Z$ is an interesting class since it cannot be larger than its complement, so it suffices to prove that $Z$ is robust.

**Lemma 9.** $Z$ is $(1 - 4c)$-robust to $L^1$-perturbations of size $c\sqrt{qh} * n - 2$

*Proof.* Let $Z' \subseteq Z$ be the set of images in $Z$ that are robust to $L^1$-perturbations of size $c\sqrt{qh} * n - 2$. Let $I$ be a random image sampled uniformly. Then $|Z'| = \Pr(I \in Z')2^{-(n^2 qh2^b)}$. We then have the following:

$$\Pr(I \in Z') = \Pr(\sum_{x,y,a} I_{x,y,a} + c\sqrt{qh} * n - 2 < n^2 qh/2) \tag{47}$$

$$\geq \Pr(\sum_{x,y,a} I_{x,y,a} \leq n^2 qh/2 - c\sqrt{qh} * n + 1) \tag{48}$$

$$> \frac{1}{2} - 2c \tag{49}$$

Where the last inequality follows from Lemma 8 since each channel is sampled from a uniform distribution over a set of $2^b$ evenly spaced values between 0 and 1. Noting that $|Z| \leq 2^{(n^2 qh2^b)-1}$ since it cannot be larger than its complement yields $\frac{|Z'|}{|Z|} \geq 1 - 4c$. Therefore, $Z$ is $(1 - 4c)$-robust to $L^1$-perturbations of size $c\sqrt{qh} * n - 2$. $\qquad \square$

**Lemma 10.** $Z$ is $(1 - 4c)$-robust to $L^0$-perturbations of size $c\sqrt{qh} * n - 2$

*Proof.* It suffices to show that an image that is robust to $L^1$-perturbations of size $d$ is also robust to $L^0$-perturbations of size $d$, since the statement then follows directly from Lemma 9.

Let $I$ be an image that is not robust to $L^0$-perturbations of size $d$, so there exists some $I'$ in a different class such that $\|I - I'\|_0 \le d$. Then:

$$d \ge \|I - I'\|_0 \tag{50}$$

$$= \sum_{(x,y,a)} \lceil |I_{x,y,a} - I'_{x,y,a}| \rceil \tag{51}$$

$$\ge \sum_{(x,y,a)} |I_{x,y,a} - I'_{x,y,a}| \tag{52}$$

$$= \|I - I'\|_1 \tag{53}$$

Where the second and third relations hold since channel values lie in $[0, 1]$.

This implies that $I$ is not robust to $L^1$-perturbations of size $d$. Therefore any image that is not robust to $L^0$-perturbations of size $d$ is also not robust to $L^1$-perturbations of size $d$. The contraposition yields the desired statement. $\square$

**Lemma 11.** *$Z$ is $(1 - 4c)$-robust to $L^p$-perturbations of size $\frac{(c\sqrt{qh}*n-2)^{1/p}}{(2^b-1)^{(p-1)/p}}$ for $p \ge 2$.*

*Proof.* It suffices to show that any image that is robust to $L^1$-perturbations of size $d$ is also robust to $L^p$-perturbations of size $\frac{d^{1/p}}{(2^b-1)^{(p-1)/p}}$ for any $p \ge 2$, since the statement then follows directly from Lemma 9.

Let $I \in \mathcal{I}_{n,q,h,(b)}$ be an image that is robust to $L^1$-perturbations of size $d$. Let $I'$ be any image in a different class, so $\|I - I'\|_0 > d$. Then for any $p \ge 1$:

$$\|I - I'\|_p^p = \sum_{(x,y,a)} |I_{x,y,a} - I'_{x,y,a}|^p \tag{54}$$

$$= \sum_{(x,y,a)} \left( \frac{(2^b - 1)|I_{x,y,a} - I'_{x,y,a}|}{(2^b - 1)} \right)^p \tag{55}$$

$$\ge \sum_{(x,y,a)} (2^b - 1)|I_{x,y,a} - I'_{x,y,a}| \left( \frac{1}{(2^b - 1)} \right)^p \tag{56}$$

$$= (2^b - 1) \frac{\|I - I'\|_1}{(2^b - 1)^p} \tag{57}$$

$$> \frac{d}{(2^b - 1)^{p-1}} \tag{58}$$

Where the second relation follows from the fact that if two channel values differ, they must differ by at least $\frac{1}{2^b-1}$.

Therefore, $\|I - I'\|_p > \frac{d^{1/p}}{(2^b-1)^{(p-1)/p}}$ for any $I'$ whose class is different from $I$, so $I$ is robust to $L^p$-perturbations of size $\frac{d^{1/p}}{(2^b-1)^{(p-1)/p}}$ for $p \ge 2$. $\square$

### A.3   PROOF OF THEOREM 3

We fix arbitrary $n$, $q$, $h$, and $b$. For simplicity, we define $N = n^2 qh$ as the data dimension. Let $C \subseteq \mathcal{I}_{n,q,h,(b)}$ be any interesting class. Our objective is to show that $C$ is not robust to various perturbations.

For the first part of the proof, we construct a grid over the unit hypercube and then map $\mathcal{I}_{n,q,h,(b)}$ to cells in this lattice while preserving distances up to a constant factor.

- Let $T$ be a set of $2^b$ disjoint intervals of equal length whose union is the interval $[0, 1]$ (specifically, we have $T = \{[x * 2^{-b}, (x + 1) * 2^{-b}) | x \in \mathbb{Z} \cap [0, 2^b - 2]\} \cup \{[1 - 2^{-b}, 1]\}$).

- Let $T^N$ be their $N$th Cartesian power. This forms a partition over the unit hypercube $[0,1]^N$ since the elements of $T^N$ are disjoint and their union is precisely the hypercube. Note that each element in the partition has equal measure.

- We can associate each element of $\mathcal{I}_{n,q,h,(b)}$ with an element of $T^N$. We first map $\mathcal{I}_{n,q,h,(b)}$ to $[0,1]^N$, which can be done by flattening the image tensor (which we denote by $\flat(I)$ for an image $I \in \mathcal{I}_{n,q,h,(b)}$). We then map that point to the element of $T^N$ the point falls within. The overall mapping is bijective, and we will denote it by $F$.

This completes the construction. To recap: each element in $\mathcal{I}_{n,q,h,(b)}$ is now associated to a cell in $T^N$ via $F : \mathcal{I}_{n,q,h,(b)} \to T^N$. $T^N$ is a partitioning of the unit hypercube into cells of equal measure.

In the next part of the proof we define an algorithm that is able to find small perturbations. We then show that this algorithm succeeds with high probability, which proves the statement.

Let $\mathcal{A} : [0,1]^N \times \mathbb{R} \to [0,1]^N \cup \{\bot\}$ be a partial function that maps a point $p_1$ and a real value $c$ to a point $p_2$ such that the following hold:

1. $\|p_1 - p_2\|_2 \leq c$.

2. Let $I_1, I_2 \in \mathcal{I}_{n,q,h,(b)}$ such that $p_1 \in F(I_1)$ and $p_2 \in F(I_2)$. Then we require that $I_1 \in C \implies I_2 \notin C$.

$\mathcal{A}(.)$ returns $\bot$ if and only if no such $p_2$ exists.

We can then define a procedure FINDPERTURBATION for finding a perturbation given an image $I$, which is outlined in Algorithm 2.

---

**Algorithm 2:** Find Perturbation

**Input :** An image $I \in C$ and a real value $c$.
**Result:** An image $I' \in \mathcal{I}_{n,q,h,(b)}$ such that $I' \notin C$, or $\bot$.
Sample $p_1$ from $F(I)$ uniformly at random;
$p_2 \leftarrow \mathcal{A}(p_1, c)$
**if** $p_2 = \bot$ **then**
  | **return** $\bot$;
**else**
  | Find $I_2 \notin C$ such that $p_2 \in F(I_2)$;
  | **return** $I_2$;

---

Our proof strategy is to show that the perturbations found by FINDPERTURBATION are guaranteed to be small, and that the probability of failure is low. This must then imply that most images are not robust.

**Lemma 12.** *If $I' = $ FINDPERTURBATION$(I, c)$ is not $\bot$, then $\|I - I'\|_2 \leq c + 2\frac{\sqrt{N}}{2^b}$.*

*Proof.* Each element of $T^N$ has a diameter of $\frac{\sqrt{N}}{2^b}$, thus $p_1$ differs from $\flat(I)$ by at most that distance. Similarly, $p_2$ differs from $\flat(I_2) = \flat(I')$ by that distance. We also must have $\|p_1 - p_2\|_2 \leq c$ since $I' \neq \bot$. Putting it altogether with the triangle inequality we get $\|\flat(I) - \flat(I')\|_2 \leq c + 2\frac{N}{2^b}$. Since $\flat(.)$ preserves distances, we get the desired statement. $\square$

**Lemma 13.** *If $I$ is drawn uniformly from $C$, then $Pr($FINDPERTURBATION$(I, c) = \bot) < 2e^{-c^2/2}$.*

*Proof.* Let $F(C)$ denote the image of $C$ under $F$. Let $\bigcup F(C)$ denote the union of all elements in $F(C)$.

If the input $I$ is drawn uniformly from $C$, then $p_1$ is distributed uniformly over $\bigcup F(C)$. The procedure fails if and only if $\mathcal{A}(p_1, c) = \bot$, which happens if and only if all elements within a radius of $c$ from $p_1$ all belong to $\bigcup F(C)$. Let $C'$ denote the set of all such points.

$$\Pr(\mathcal{A}(I_2, c) = \bot) = \frac{\mu(C')}{\mu(\bigcup F(C))} \tag{59}$$

$$< 2e^{-c^2/2} \tag{60}$$

Where $\mu(.)$ denotes the Lebesgue measure.

The last inequality comes from Theorem 5, which is given in the next section. The statement applies for any set $S$ formed from a union of elements of $T^N$ whose measure is no larger than $1/2$. $\bigcup F(C)$ satisfies these criteria since $C$ is an interesting class and each element of $T^N$ is of equal measure and disjoint, so we attain the desired statement. $\qquad\square$

**Lemma 14.** $C$ is not $2e^{-c^2/2}$-robust to $L^2$-perturbations of size $c + 2\frac{\sqrt{N}}{2^b}$.

*Proof.* Let $I$ be drawn uniformly from $C$. Let $C_r$ be the set of images that are robust to $L^2$-perturbations of size $c + 2\frac{\sqrt{N}}{2^b}$.

Let $I' = \text{FINDPERTURBATION}(I, c)$. Then $I'$ is randomly distributed over $\mathcal{I}_{n,q,h,(b)} \cup \{\bot\}$. By Lemma 12, if $I' \in \mathcal{I}_{n,q,h,(b)}$, then $\|I - I'\|_2 \leq c + 2\frac{\sqrt{N}}{2^b}$, which implies that $I \notin C_r$. By contraposition, $I \in C_r$ implies that $\text{FINDPERTURBATION}(I, c) = \bot$. Therefore:

$$\Pr(I' = \bot) = \Pr(I \in C_r) + \Pr(I \notin C_r, I' = \bot) \tag{61}$$

$$\geq \Pr(I \in C_r) \tag{62}$$

$$= \frac{|C_r|}{|C|} \tag{63}$$

By Lemma 13, $\Pr(I' = \bot) < 2e^{-c^2/2}$. Thus, $\frac{|C_r|}{|C|} < 2e^{-c^2/2}$, which yields the desired statement. $\qquad\square$

**Lemma 15.** $C$ is not $2e^{-c^2/2}$-robust to $L^p$-perturbations of size $\left(c + 2\frac{\sqrt{N}}{2^b}\right)^{2/p}$ for $p \geq 2$.

*Proof.* We use the identical argument from Lemma 6.

Let $S_1$ be the set of images that are $r$-robust to $L^2$-perturbations of size $d$, and let $S_2$ be the set of images that are $r$-robust to $L^p$-perturbations of size $d^{2/p}$, where $p \geq 2$.

Suppose $I \notin S_1$. Then there exists some image $I'$ in a different class from $I$ such that $\|I - I'\|_2 \leq d$. Therefore, for all $p \geq 2$, we have:

$$d^2 \geq \|I - I'\|_2^2 \tag{64}$$

$$= \sum_{x,y,c} |I_{x,y,c} - I'_{x,y,c}|^2 \tag{65}$$

$$\geq \sum_{x,y,c} |I_{x,y,c} - I'_{x,y,c}|^p \tag{66}$$

$$= (\|I - I'\|_p)^p \tag{67}$$

Where the third relation follows from the fact that channel values are contained in $[0, 1]$. Therefore, $I \notin S_2$ either since $\|I - I'\|_p \leq d^{2/p}$. Taking the contraposition yields $S_2 \subseteq S_1$.

Setting $d = c + 2\frac{\sqrt{N}}{2^b}$ and applying Lemma 14 gives the desired result. $\qquad\square$

Substituting $\sqrt{N} = \sqrt{qh} * n$ to Lemma 6 yields the exact form of Theorem 3.

## A.4 PROOF OF THEOREM 5

Our objective in this section is to complete the proof of Theorem 3 by proving Theorem 5, stated below. We will use $\mu(.)$ to denote Lebesgue measure throughout this section.

**Definition 4.** *We say a set $S \subseteq [0,1]^n$ is a regular set if there exists some finite set $T$ such that $S = \bigcup_{t \in T} t$ and $T$ consists of elements that are Cartesian products of $n$ intervals that are either open or closed.*

By this definition the sets defined in the proof of Lemma 13 are regular, so the following theorem is applicable.

**Theorem 5.** *Let $S \subseteq [0,1]^n$ be a regular set such that $\mu(S) \leq 1/2$. Let $S_r \subseteq S$ contain all the points in $S$ such that for all $y \in [0,1]$, $\|x - y\|_2 \leq r \implies y \in S$. Then $\frac{\mu(S_r)}{\mu(S)} < 2e^{c^2/2}$.*

### A.4.1 PROPERTIES OF THE STANDARD NORMAL DISTRIBUTION

First, we define the cumulative distribution function for the standard normal distribution and its derivative.

$$\Phi(x) = \int_{-\infty}^{x} \frac{1}{\sqrt{2\pi}} e^{-t^2/2} dt \tag{68}$$

$$\Phi'(x) = \frac{1}{\sqrt{2\pi}} e^{-x^2/2} \tag{69}$$

Similarly to the discrete case, the ratio of the cumulative distribution functions is monotonic increasing.

**Lemma 16.** $\frac{\Phi(x-k)}{\Phi(x)}$ *is monotonic increasing in $x$ for all $k \geq 0$.*

*Proof.* Let $f(x) = \frac{e^{-x^2/2}}{\int_{-\infty}^{x} e^{-t^2/2} dt}$. Then:

$$\frac{d}{dx} f(x) = \frac{-e^{-x^2/2} x \int_{-\infty}^{x} e^{-t^2/2} dt - e^{-x^2/2} e^{-x^2/2}}{\left( \int_{-\infty}^{x} e^{t^2/2} dt \right)^2} \tag{70}$$

When $x \geq 0$, this derivative is negative since both terms in the numerator are negative. If $x < 0$, we have the following:

$$-x \int_{-\infty}^{x} e^{-t^2/2} dt < -x \int_{-\infty}^{x} e^{-t^2/2} + \frac{1}{t^2} e^{-t^2/2} dt \tag{71}$$

$$= -x \left( -\frac{1}{t} e^{-t^2/2} \Big|_{-\infty}^{x} \right) \tag{72}$$

$$= e^{-x^2/2} \tag{73}$$

So the sum is strictly smaller than $(e^{-x^2/2})^2 - (e^{-x^2/2})^2 = 0$. Therefore, the derivative is everywhere negative, so $f(x)$ is strictly decreasing.

Therefore, we have the following for any non-negative $k$:

$$\frac{d}{dx} ln(\frac{\Phi(x-k)}{\Phi(x)}) = f(x-k) - f(x) \geq 0 \tag{74}$$

Since $ln(.)$ is a monotonic increasing function, $\frac{\Phi(x-k)}{\Phi(x)}$ must also be monotonic increasing. □

### A.4.2 PROVING THEOREM 5

Similarly to the discrete case, our main result relies on an isoperimetry statement, this time on the unit hypercube (Barthe & Maurey, 2000).

**Lemma 17** (Isoperimetric Theorem on the Unit Hypercube). *For any $n$, let $A \subset [0,1]^n$ be a Borel set. Let $A_\epsilon = \{x \in [0,1]^n | \exists x' \in A : \|x - x'\| \leq \epsilon\}$. Then we have the following:*

$$\liminf_{\epsilon \to 0^+} \frac{\mu(A_\epsilon) - \mu(A)}{\epsilon} \geq \sqrt{2\pi} \Phi'(\Phi^{-1}(\mu(A))) \tag{75}$$

Let $C \subseteq [0,1]$ be a regular set such that $0 < \mu(C) \leq 1/2$. Let $C_r \subseteq C$ denote the points $p_1$ in $C$ such that for any point $p_2 \in [0,1]$, $\|p_1 - p_2\|_2 \leq r \implies p_2 \in C$.

**Lemma 18.** $C_r \leq \Phi(\Phi^{-1}(\mu(C)) - r)$

*Proof.* Let $z = \Phi^{-1}(\mu(C))$ and let $f(x) = \Phi(x + z)$. Let $v(.)$ be a Lebesgue integrable function such that the following holds:

$$V(r) = \int_{(-\infty, r)} v(t)dt = \begin{cases} \mu(C_{-r}) & \text{if } r \leq 0 \\ \mu(C_0) & \text{otherwise} \end{cases} \tag{76}$$

This exists since $C$ is a regular set. Since $V(x)$ results from integration, it is also a continuous function.

It then suffices to show that $V(x) \leq f(x)$ for all $x$, since $V(x)$ corresponds to the left hand side of the theorem statement and $f(x)$ corresponds to the right hand side. Suppose this is not the case. We know that $V(x) \leq f(x)$ for all $x \geq 0$, so if this is violated it must happen when $x < 0$. Since $V(x)$ and $f(x)$ are both continuous, by the intermediate value theorem there must exist some interval $[a, b)$ where $V(x) > f(x)$ if $x \in [a, b)$, $V(b) = f(b)$, and $a < b \leq 0$.

This gives us the following:

$$V(b) - V(a) = \int_{[a,b)} v(t)dt \tag{77}$$

$$= \int_{[a,b) \setminus Z} \lim_{\epsilon \to 0^+} \frac{V(t + \epsilon) - V(t)}{\epsilon} dt \tag{78}$$

$$= \int_{[a,b) \setminus Z} \liminf_{\epsilon \to 0^+} \frac{\mu(C_{-t-\epsilon}) - \mu(C_{-t})}{\epsilon} dt \tag{79}$$

$$\geq \int_{[a,b)} \sqrt{2\pi} \Phi'(\Phi^{-1}(\mu(C_{-t}))) dt \tag{80}$$

$$\geq \int_{[a,b)} \sqrt{2\pi} \Phi'(\Phi^{-1}(f(t))) dt \tag{81}$$

$$\geq f(b) - f(a) \tag{82}$$

Where $Z$ is the set of values where the limit in Equation 78 is not equal to $v(t)$, which by the Lebesgue differentiation theorem is a set of measure 0. Equation 80 is an application of Lemma 17, which is applicable since $C_{-t}$ is a Borel set because $C$ is a regular set. Equation 81 follows from the fact that $f(x) \leq V(x)$ for all $x \in [a, b]$ and the fact that $\Phi'(\Phi^{-1}(.))$ is monotonically increasing if the input is no greater than $1/2$.

We also have $V(a) > f(a)$ and $V(b) = f(b)$, so it must be the case that $V(b) - V(a) < f(b) - f(a)$. This contradicts the above, so it must be the case that $V(x) \leq f(x)$ for all $x$. $\quad\square$

**Lemma 19.** $\mu(C_c) < 2e^{-c^2/2}\mu(C)$

*Proof.* Let $z = \Phi^{-1}(\mu(C))$. Then for any $c \geq 0$,

$$\frac{\mu(C_c)}{\mu(C)} \leq \frac{\Phi(z-c)}{\Phi(z)} \leq \frac{\Phi(1/2-c)}{\Phi(1/2)} < 2e^{-c^2/2} \tag{83}$$

Where the first inequality follows from Lemma 18, the second inequality follows from Lemma 16 and the fact that $\mu(C) \leq 1/2$, and the third inequality follows from the Gaussian tail bound $\Phi(x) < e^{-x^2/2}$ for all $x \leq 1/2$. $\qquad\square$

## A.5 AVERAGE DISTANCE BETWEEN IMAGES

We wish to show that for a pair of images $I, I' \in \mathcal{I}_{n,q,h,(b)}$ that are sampled independently and uniformly, there exists a $k_{b,p}$ such that:

$$\mathbb{E}[\|I - I'\|_p] \geq k_{b,p} N^{1/\max(1,p)} \tag{84}$$

If $p = 0$ then set $k_{b,p}$ to $1 - 2^{-b}$ and the relation will hold with equality, so we are done. Otherwise, we note that we have:

$$\mathbb{E}[\|I - I'\|_p^{\max(1,p)}] = N * \mathbb{E}[|X - Y|^{\max(1,p)}] \tag{85}$$

Where $X$ and $Y$ are independent random variables that are both drawn uniformly from a set of $2^b$ equally spaced values, where the largest is 1 and the smallest is 0. For simplicity, we denote $\mathbb{E}[|X - Y|^{\max(1,p)}]$ with $z$.

$\|I - I'\|_p^{\max(1,p)}$ is non-negative and cannot be larger than $N$. Therefore, the probability that $\|I - I'\|_p^{\max(1,p)} \geq Nz/2$ is at least $\frac{z}{2-z}$.

Via a monotonicity argument we can deduce that the probability that $\|I - I'\|_p \geq (z/2)^{1/\max(p,1)} N^{1/\max(p,1)}$ is at least $\frac{z}{2-z}$ as well. We can then apply Markov's inequality to get the following:

$$\mathbb{E}[\|I - I'\|_p] \geq \frac{z}{2-z} (z/2)^{1/\max(p,1)} N^{1/\max(p,1)} \tag{86}$$

By setting $k_{b,p}$ to be $\frac{z}{2-z}(z/2)^{1/\max(p,1)}$ we attain our desired result.

### A.5.1 AVERAGE DISTANCES BETWEEN IMAGES FROM NATURAL DISTRIBUTIONS ARE ALSO LARGE

The above analysis shows that average distances between images over the entire image space is large, but it does not preclude the possibility that average distances between images from a distribution of natural images is small.

To investigate this, we computed average distances between images in Imagenette, a subset of Imagenet consisting of 10 classes (tench, English springer, cassette player, chain saw, church, French horn, garbage truck, gas pump, golf ball, parachute)(Howard). In detail, we took all the images within the training set of Imagenette with the shorter side resized to 320 pixels and discarded all images that were not in RGB, resulting in 9296 images. Each image had a bit depth of 8. We then cropped the images such that only the top left 320x320 pixels remained. We then computed the average distance measured in $p$-norms where $p$ ranged from 1 to 5:

$$\frac{\sum_{x \in D} \sum_{y \in D} \|x - y\|_p}{|D|^2} \tag{87}$$

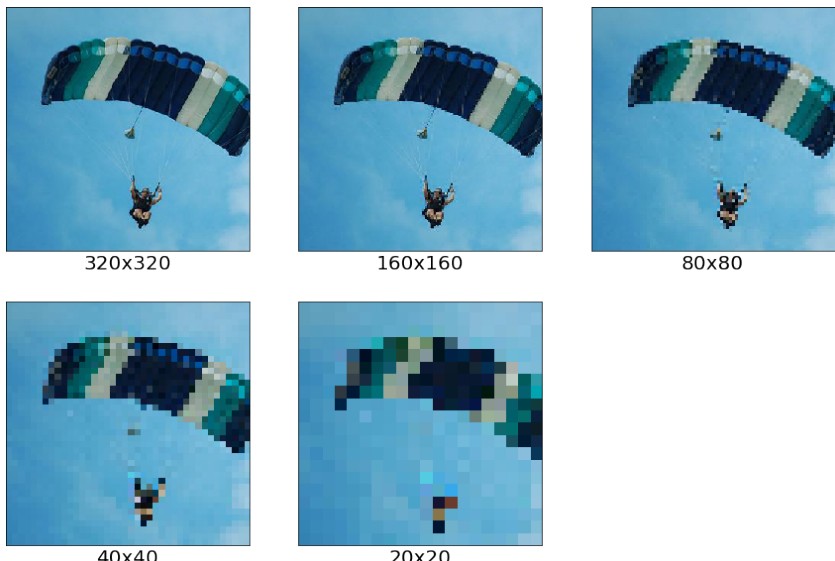

Figure 3: Subsampling 320x320 images to smaller sizes

Where $D$ is the set of 320x320 images. We also subsampled the images to 160x160, 80x80, 40x40, and 20x20 by taking the top left pixel of each region as the representative sample (see Figure 3) and computed the average distances between them too. The results are presented in Figure 4, and indicate that the average distances between images drawn from natural distributions are not dramatically lower than those between images drawn uniformly if we believe the Imagenette dataset is sufficiently representative of natural image distributions.

The average distances between uniformly drawn images were each approximated with 200000 pairs of randomly drawn images, except for the 1-norm where a closed form solution exists: for a pair of $n$x$n$ images with $h$ channels and bit depth $b$, the average distance between all pairs of images, measured with the 1-norm, is $c_b n^2 h$, where $c_b$ is given by:

$$c_b = \frac{\sum_{i=0}^{2^b-1} \sum_{j=0}^{2^b-1} \frac{|i-j|}{2^b-1}}{2^{2b}} \tag{88}$$

## A.6 PLOTS OF THE RELATION BETWEEN ROBUSTNESS AND PERTURBATION SIZE

We plotted the curves defined by the upper bounds in Table 1 to facilitate interpretation. The curves are plotted in Figure 5. $h$, $b$, and $q$ have been fixed at 3, 8, and 1 respectively, so the bounds apply to square RGB images where each channel has a bit depth of 8. Perturbations sizes are then plotted on the $y$-axis, and a corresponding upper bound on the robustness achievable is plotted on the $x$-axis on a logarithmic scale. Sizes are measured in 0-norms, 1-norms, and 2-norms, and we plot these curves for $n = 32$, $n = 256$, and $n = 1024$.

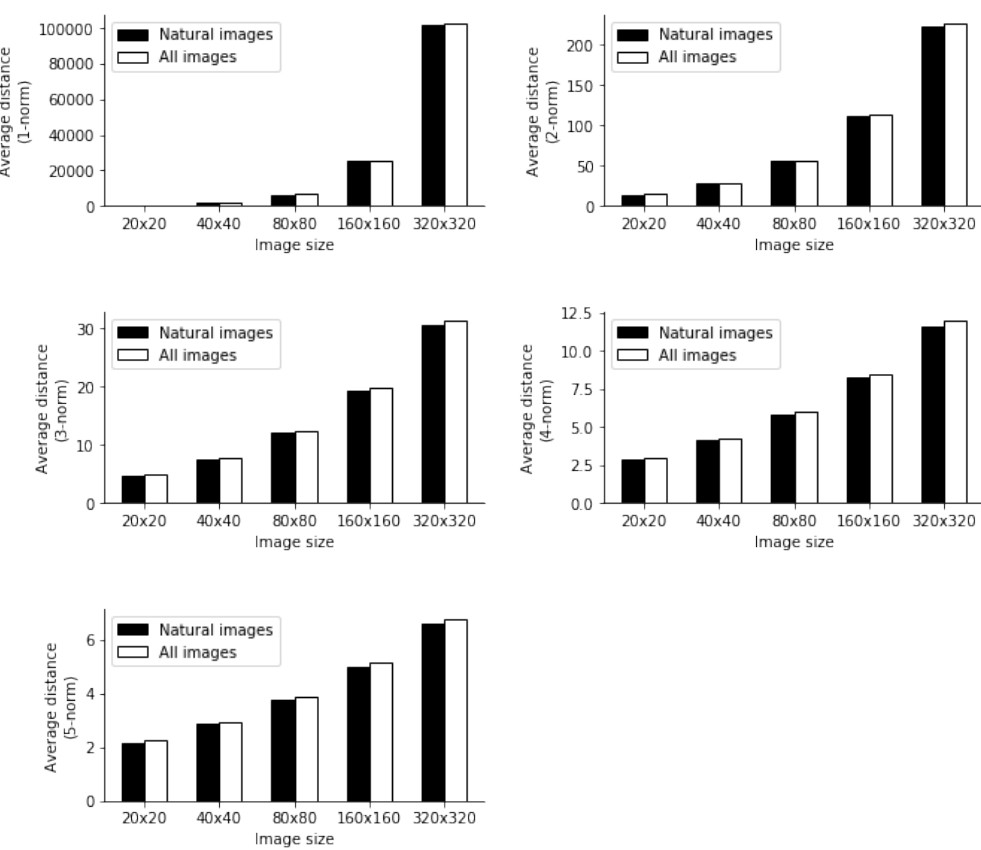

Figure 4: Comparison of average distances between images from natural distributions with average distances between uniformly drawn images on various $p$-norms

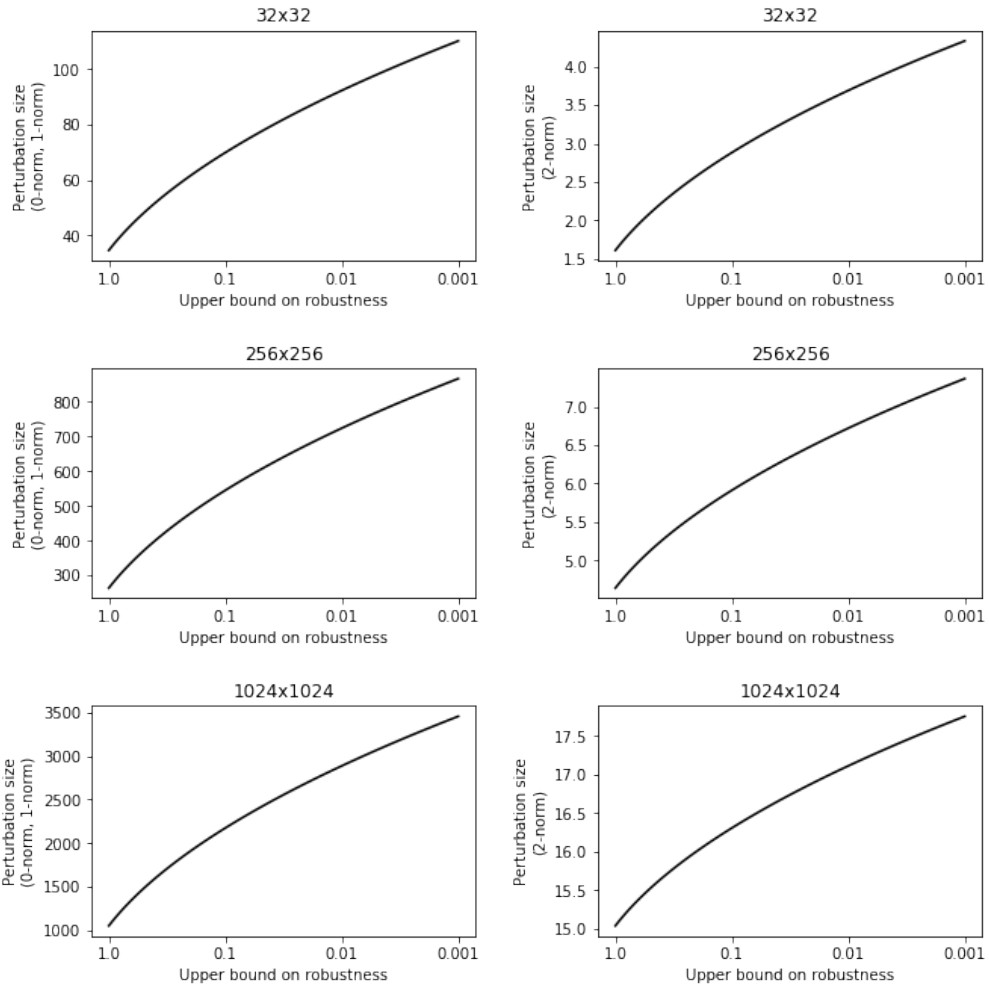

Figure 5: Relation between perturbation sizes and corresponding upper bounds on achievable robustness of interesting classes.

# B APPENDIX

## B.1 ADAPTING OF LEMMA 3 FROM (HARPER, 1999)

In this section we will show how we adapt Theorem 3 from (Harper, 1999) in its exact form into the form given in Lemma 3.

### B.1.1 PRELIMINARIES

Some additional terminology is required to parse Theorem 3 from (Harper, 1999) in its original form. We note that the terminology concern elements from the unit hypercube $[0, 1]^d$.

$H(x, y)$ is used to denote the Hamming distance between two elements $x, y \in [0, 1]^d$, in other words the number of coordinates at which they differ.

A *lower set* is a set $S \subseteq [0, 1]^d$ such that if $x \in S$, then any vector $y \in [0, 1]^d$ with the property $y_i \leq x_i$ for all $i$ where $1 \leq i \leq d$ is also in $S$.

A weighting $t$ is a vector in $(0, 1)^d$ that splits the unit hypercube $[0, 1]^d$ into $2^d$ distinct regions. The set of regions is $\times_{i=1}^d \{[0, t_i], (t_i, 1]\}$, and it is said that $t$ is constant if $t_i = t_j$ for all $i, j$ where $1 \leq i \leq d$ and $1 \leq j \leq d$. Here we use $\times$ to denote the Cartesian product.

$HB(r, d)$ with weighting $t$ is a subset of $[0, 1]^d$, and is referred to as a *Hamming ball* of radius $r$. A Hamming ball of radius $r$ is usually used to denote a subset of $\{0, 1\}^d$ that contains all vectors where the sum of the entries does not exceed $r$. This concept can be naturally be extended to $[0, 1]^d$ with weighting $t$: if we define $F_i$ as a map that maps 0 to $[0, t_i]$ and 1 to $(t_i, 1]$, we have:

$$HB(r, d) = \bigcup_{v \in H_r} \times_{i=1}^d F_i(v_i) \tag{89}$$

Where $H_r$ is a Hamming ball of radius $r$ in the traditional sense.

### B.1.2 ADAPTING THE NOTATION

We are now ready to give Theorem 3 as it is stated in (Harper, 1999).

**Theorem.** $\forall v, 0 < v < 1, \exists r$ *such that* $HB(r, d)$ *minimizes*

$$|\Phi_h(r, d)| = |\{y \notin S : \exists x \in S, H(x, y) \leq h\}|$$

*over all lower sets,* $S \subseteq [0, 1]^d$, *with* $|S| = v$. *The optimal weighting for* $HB(r, d)$ *is constant*

To bring the theorem statement closer to our terminology, we have $\text{EXP}^h(S) = S \cup \{y \notin S : \exists x \in S, H(x, y) \leq h\}$. Since the volume of $S$ is constant, $HB(r, d)$ also minimizes $|\text{EXP}^h(S)|$ over all lower sets $S$ of a given volume. Note that for our expansion notation to make sense here, we act as though $[0, 1]^d$ is an infinite graph where there exists an edge between any pair of vectors that differ at exactly one index.

Since the theorem states that the weighting $t$ should be constant, we have an exact form for $HB(r, d)$. Let $p$ denote the value of all entires of $t$. We then have $|HB(r, d)| = U_{d,p}(r)$ and $|\text{EXP}^h(HB(r, d))| = U_{d,p}(r + h)$.

Therefore, for any lower set $S$, it must be the case that

$$|\text{EXP}^h(S)| \geq \min\{|\text{EXP}^h(HB(r, d))| \mid t \in (0, 1)^d, r \in [0, d - h), |HB(r, d)| = |S|\} \tag{90}$$

$$= \min\{U_{d,p}(r + h) \mid r \in [0, d - h), p \in (0, 1), U_{d,p}(r) = |S|\} \tag{91}$$

We can then restate Theorem 3 from (Harper, 1999) in the following form:

**Lemma 20.** *Let $S \subsetneq [0,1]^n$ be a lower set. Then:*

$$|\text{Exp}^k(S)| \geq \min\{U_{n,p}(r+k)$$
$$|U_{n,p}(r) = |S|,$$
$$p \in (0,1), r \in [0, n-k]\}$$

Note that we have replaced $d$ with $n$ and $h$ with $k$ to match our notation. This is now nearly in the form of Lemma 3. All that remains is to show that this inequality implies an analogous one for Hamming graphs.

### B.1.3 EMBEDDING A HAMMING GRAPH

Let $[w]$ be the set integers from 1 to $w$ inclusive and let $\mathcal{H}(n,w)$ be the Hamming graph with the vertex set $[w]^n$. If $S \subseteq [w]^n$, we call it a lower set if $x \in S$ implies that any $y$ with the property that $y_i \leq x_i$ for all $i$ where $1 \leq i \leq n$ is also in $S$.

**Lemma 21.** *For all $S \subseteq [w]^n$, there exists a lower set $S'$ where $|S'| = |S|$ and $|\text{Exp}^k(S')| \leq |\text{Exp}^k(S)|$.*

*Proof.* Let $S \subseteq [w]^n$. Let $i$ be an index such that there exists some $x \in S$ and $y \notin S$ such that $y_i < x_i$, and for all $j$ $y_j \leq x_j$. If not such $i$ exists, then we are done, since $S$ must be a lower set.

Group together all vertices that are equal on all coordinates except $i$. Each group contains exactly $w$ elements. For each group $G$, reorganize which elements belong to $S$ by moving them to the bottom. Denote the new set by $S^*$. For example, suppose that $G \cap S = \{(1,2,2), (1,2,3), (1,2,5)\}$, and $i = 3$. Then we will shuffle those elements downwards such that we get $G \cap S^* = \{(1,2,1), (1,2,2), (1,2,3)\}$.

Note that this action cannot increase the size of the expansion. Given a group $G$, consider the number of elements that are not in $\text{Exp}^k(S)$. Denote it by $z$. If $z = 0$, then that number clearly cannot decrease after reorganization.

If $z \neq 0$, then no vertex that is within $k$ steps of any element of $G$ is a member of a group $G'$ such that $|G' \cap S| > w - z$. Otherwise, $G$ must have more than $w - z$ elements that can be reached by those elements in $k$ steps, which means $G$ must have less than $z$ elements that are not in $\text{Exp}^k(S)$.

But then that means $\text{Exp}^k(S^*)$ cannot reach more than $w - z$ elements of $G$, specifically the ones where the $i$th coordinate is at most $w - z$. Therefore, $G$ has at least $z$ elements that are not in $\text{Exp}^k(S^*)$. Therefore $|\text{Exp}^k(S^* \cap G)| \leq |\text{Exp}^k(S \cap G)|$, and since every vertex is covered by exactly one group, we have $|\text{Exp}^k(S^*)| \leq |\text{Exp}^k(S)|$.

We can keep iterating this. With each iteration, the sum of all coordinates of all elements in $S$ will strictly decrease, and since this sum is positive, this process will eventually terminate, leaving us with a lower set $S'$ that has equal size and with an expansion that is no larger than that of the original set. $\square$

We can map elements from $\mathcal{H}(n,w)$ to subsets of $[0,1]^n$ via the following map:

$$F((x_1, x_2, ..., x_n)) = [\frac{x_1 - 1}{w}, \frac{x_1}{w}] \times [\frac{x_2 - 1}{w}, \frac{x_2}{w}] \times ... \times [\frac{x_n - 1}{w}, \frac{x_n}{w}] \tag{92}$$

This mapping preserves expansion for any set $S$ in the following sense:

$$\bigcup_{x \in \text{Exp}^k(S)} F(x) = \text{Exp}^k(\bigcup_{x \in S} F(x)) \tag{93}$$

Furthermore, we have for any set $S$:

$$\frac{|S|}{|V(\mathcal{H}(n,q))|} = |\bigcup_{x \in S} F(x)| \tag{94}$$

Finally, this mapping also preserves the property of being a lower set.

Therefore, for any set $S \subseteq \mathcal{H}(n, w)$, we have:

$$\frac{|\text{Exp}^k(S)|}{|V(\mathcal{H}(n, w))|} \geq \min\{\frac{|\text{Exp}^k(S')|}{|V(\mathcal{H}(n, w))|} \, |S' \subseteq \mathcal{H}(n, w) \text{ is a lower set}, |S'| = |S|\} \tag{95}$$

$$= \min\{|\bigcup_{x \in \text{Exp}^k(S')} F(x)| \, |S' \subseteq \mathcal{H}(n, w) \text{ is a lower set}, |S'| = |S|\} \tag{96}$$

$$= \min\{|\text{Exp}^k(\bigcup_{x \in S'} F(x))| \, |S' \subseteq \mathcal{H}(n, w) \text{ is a lower set}, |S'| = |S|\} \tag{97}$$

$$\geq \min\{|\text{Exp}^k(S')| \, |S' \subseteq [0, 1]^n \text{ is a lower set}, |S'| = \frac{|S|}{|V(\mathcal{H}(n, w))|}\} \tag{98}$$

$$\geq \min\{U_{n,p}(r + k) \, |U_{n,p}(r) = \frac{|S|}{|V(\mathcal{H}(n, w))|}, p \in (0, 1), r \in [0, n - k]\} \tag{99}$$

The second and third relations are properties of the mapping $F$, the fourth relation holds because we are expanding the set of sets that we consider, and the last relation follows from our derivation in the previous section.

The gives the statement of Lemma 3, which we restate here for convenience:

**Lemma** (Isoperimetric Theorem on Hamming graphs). *Let $S \subsetneq \mathcal{H}(n, w)$. Then:*

$$\frac{|\text{Exp}^k(S)|}{|V(\mathcal{H}(n, w))|} \geq \min\{U_{n,p}(r + k)$$

$$|U_{n,p}(r) = \frac{|S|}{|V(\mathcal{H}(n, w))|},$$

$$p \in (0, 1), r \in [0, n - k]\}$$

