# OpenReview forum: "Fundamental limits on the robustness of image classifiers"
_ICLR.cc/2023/Conference — ICLR 2023 poster_

### Official Review · Reviewer_sETu · 2022-10-13

**Confidence:** 4
**Correctness:** 4
**Technical Novelty And Significance:** 3
**Empirical Novelty And Significance:** 3
**Recommendation:** 8

**Clarity, Quality, Novelty And Reproducibility:**

Clarity: very good. I really enjoyed reading the whole paper, especially the abstract. If possible, maybe the notations in the proof of Theorem 3 can be simplified. I find some of the notations in the proof is not so familiar to me.

Quality: very good. The proof is clear and rigorous.

Novelty: very good. As I said, this is something very good to know, but previously I personally never heard of this.

Reproducibility: good. It's a theory paper, with a very intuitive main theorem. I believe this theorem is correct.


**Strength And Weaknesses:**

For a long time, many people talk about the origin of adversarial examples. Some people believe this is due to the inherent structural problems in neural networks -- this might be true, but this paper provides a completely different perspective. This paper proves that, no matter which algorithm we are using, we will always get adversarial examples. The adversarial examples are the products of the high dimensional image space.

This gives a very nice lower bound of robustness theory, just like the no free lunch theorem for Generalization. Similarly, I think this is not to say that we will not get anything robust for real problems, that is too pessimistic -- just like no free lunch theorem does not imply that we will not get anything learnable. Instead, it tells us that we should be more careful about the notion "robustness", e.g., with additional priors or assumptions, otherwise nothing is robust.

I think the paper can be further improved with the following two aspects:

1. Can the authors describe the relationship/gap between their main theorem and the actual real data distribution? At least from my own perspective, for datasets like CIFAR-10/Imagenet, if there are M classes of images, the total mass of all the classes is less than half of the space. The most points, belonging to the uninteresting class according to Definition 1 in the paper, can be seen as some noisy/weird points. In that case, it seems that the main theorem is still true, but after perturbation, the perturbed points will move from certain interesting class to an uninteresting class. In practice, what we observe is different: these points will move to other interesting classes. To me, it seems that designing a good model that can incorporate the notion of uninteresting class does not violate the main theorem, but is very useful in practice. Can the authors comment on this?

2. Can the techniques for bit-depth also applied to the size of the image and channels? It seems to me that all the four dimensions are symmetric, so why is bit-depth so special here?



**Summary Of The Paper:**

This paper proves an isoperimetric theorem for the image space. Specifically, it shows, in the pixel space of images, for every class of size less than half of the space, most of the points in the class are located on the boundary of the class (hence a small adversarial perturbation will move the point out of the class).

This claim is not surprising in high dimensional space (in fact, it is pretty intuitive). Their proof is also based on the existing result on Hamming graphs. However, I think this is a very interesting paper, because in retrospect, I am surprising that people previously did not realize this simple fact. In my opinion, this is like the “no free lunch” theorem in the Generalization theory, which gives a very simple but important lower bound of what can be learned and what cannot.

(The paper also has a few additional results, including the bounds they have are tight by giving one specific synthetic example, and also improvement of the bound when adjusting the bit depth of the image space)

Therefore, I think this paper is a clear accept.


**Summary Of The Review:**

This paper provides a very simple but important lower bound argument of the robustness in image space, based on the isoperimetric theorem in Hamming graphs. The result is very good to know, for people doing research in robustness ML.

---

> ### Author Response · Authors · 2022-11-15
> **Response to reviewer**
>
> We thank the reviewer for their complimentary review, and for their thoughtful and insightful comments. We are very glad that the reviewer appreciates the value of our work and its potential to inform future research. We would like to respond to the following comments and questions by the reviewer:
>
> *“Can the authors describe the relationship/gap between their main theorem and the actual real data distribution?”*
>
> One way of capturing the relationship/gap between our theoretical results and the main theorem is through $\omega$ described in Section 3.2. However, quantifying $\omega$ is a tricky task since it requires a good understanding of the real data distribution, and without additional priors there is no non-trivial theoretical bounds we can impose on it as demonstrated by the toy example we introduce along with $\omega$.
>
> Separately, we note that typical distances between images from the entire image space and typical distances between images from some real distribution look similar, at least empirically, so the bounds we derive are small even when compared to the typical distances of some real distribution. This comparison is made in the newly added Appendix A.5.1, where the evaluation is done using Imagenette (a subset of Imagenet).
>
> *“At least from my own perspective, for datasets like CIFAR-10/Imagenet, if there are M classes of images, the total mass of all the classes is less than half of the space. The most points, belonging to the uninteresting class according to Definition 1 in the paper, can be seen as some noisy/weird points. In that case, it seems that the main theorem is still true, but after perturbation, the perturbed points will move from certain interesting class to an uninteresting class. In practice, what we observe is different: these points will move to other interesting classes. To me, it seems that designing a good model that can incorporate the notion of uninteresting class does not violate the main theorem, but is very useful in practice. Can the authors comment on this?”*
>
> The reviewer’s observations are correct: while the main theorem itself does not guarantee what the class changes under small perturbations will be (only that there will be a class change), we can at least conclude that many of these perturbations would be towards uninteresting images by noting that the total mass of interesting classes such as those in CIFAR-10/Imagenet should constitute less than half the space.
>
> If we understand the reviewer correctly, there’s an implicit observation that given a small perturbation, a point moving from an interesting class to an uninteresting one is perhaps better behavior than a point moving from an interesting class to another interesting class. Since the main theorem indicates that at least one of these should be the case, it may be helpful to explicitly construct a classifier that would behave like the former, perhaps by way of introducing an explicit uninteresting class.
>
> This is an interesting idea. The difficulty here is implementation - there are many ways that inputs can be “uninteresting”. The ability to construct an uninteresting class essentially provides the means of producing an out of distribution detector, which is an active area of research. It could indeed be the case that composing an out of distribution detector with a classifier is a viable way of creating a good model in practice.
>
> *“Can the techniques for bit-depth also applied to the size of the image and channels? It seems to me that all the four dimensions are symmetric, so why is bit-depth so special here?”*
>
> We wish to clarify that $I_{n,q,h,b}$ is actually a set of 3 dimensional tensors of shape $(qn, n, h)$. $b$ does not affect the shape of the tensors, instead it parametrizes how dense the images are, so $b$ is special. We apologize for the confusing notation, which we have now updated to $I_{n,q,h,(b)}$ to emphasize the special role played by $b$.
>
> *“If possible, maybe the notations in the proof of Theorem 3 can be simplified.”*
>
> We also thank the reviewer for pointing out the complexity in the proof of Theorem 3. We have simplified some of the notation and have revised it with additional commentary to make the proof clearer (Appendix A.3).

---

### Official Review · Reviewer_xoPQ · 2022-10-22

**Confidence:** 3
**Correctness:** 4
**Technical Novelty And Significance:** 3
**Empirical Novelty And Significance:** Not applicable
**Recommendation:** 6

**Clarity, Quality, Novelty And Reproducibility:**

Clarity: the paper is well-written and reasonably clear to follow. The discussion on semantically salient features seems to depart from the largely theoretical theme of the paper however.

Novelty: The proof techniques are novel and interesting. Robustness impossibility based on isoperimetry-like arguments is not too novel, but the discrete setting is satisfying.

Reproducibility: N/A -- no real empirical results

**Strength And Weaknesses:**

Strengths:
Adversarial robustness is an important and busy area of study. This work offers some insight into the impossibility of robustness of robustness for discrete-valued vector functions, using a novel proof technique. In particular, the consideration of finite bit-depth in the results in this work are interesting and novel. The theoretical results presented require very few assumptions and can generalize to all classes of functions. Further, it is quite appealing to see that upper bounds can be made tight(ish) by matching lower bounds.

Weaknesses:
My only complaint with this line of work is that it may be interpreted as offering explanation for the inability for images from the data distribution (e.g. cifar10, imagenet) to be robust to Lp-bounded adversaries. While the theoretical results are sound, they do not necessarily speak to the applications of interest with respect to robustness. For example, consider all the images labeled 'cat' in the CIFAR10 dataset $X_{cat}$, and consider the preimage of the label 'cat' for some ideal classifier $\cal{Y} = f^{-1}(\text{cat})$. It is entirely possible that $\cal{Y}$ is an interesting class, while simultaneously containing the union of $L_p$ balls for every $x\in X_{cat}$: i.e. $\bigcup_{x\in X_{cat}} B(x, \epsilon) \subseteq \cal{Y}$. While the results in Theorem 1 would still hold for the interesting class $\cal{Y}$, one might also say that the network $f$ is robust to an Lp bounded adversary for the cat class. In this sense, while the proof techniques are novel and interesting, the results seem natural from isoperimetric arguments. Thus, it's unclear how applicable such results as these are to building the intuition of robustness researchers at large, given that it's generally believed that images live on a lower-dimensional manifold (and therefore have measure 0 in ambient space). Nevertheless, I think this complaint is relatively minor in light of the theoretical results provided, perhaps just a bit of overselling occurring by arguing this applies more broadly to adversarial robustness in-practice.

**Summary Of The Paper:**

The authors present theoretical results on the robustness of image classifiers. Namely they demonstrate that for all image classifiers, the set of images robust to Lp-bounded adversarial perturbations becomes vanishingly small. The main technique used in the proofs is a novel approach leveraging Hamming graphs. The bounds provided are asymptotically optimal, up to constants. Some discussion human perception is also provided.

**Summary Of The Review:**

Thiis paper provides novel theoretical results on the limits of robustness for a broad class of functions. The proofs provide matching(ish) lower and upper bounds, and are novel in their technique as well as investigation of bit-depth. However, the generality and nature of high-dimensional spaces do not lend themselves to providing results that seem to be applicable for many real-world cases where robustness is only considered as an epsilon-dilation of a lower dimensional manifold. Overall, I would like to see this paper accepted, however for the aforementioned reasons, I do not feel very strongly about it and am generally unwilling to work too hard convincing other reviewers in arguing for its acceptance.

---

> ### Author Response · Authors · 2022-11-15
> **Response to reviewer**
>
> We thank the reviewer for their thoughtful comments and favorable review. We are glad that the reviewer sees the theoretical value in this work. We would like to remark on the following comments made by the reviewer:
>
> *“While the results in Theorem 1 would still hold for the interesting class $\mathcal{Y}$, one might also say that the network $f$ is robust to an Lp bounded adversary for the cat class”*
>
> In essence, the reviewer points out that robustness can still be achieved when we only consider important subsets of the image space, such as the natural image manifold. We agree with this observation, and we did note in Section 3.2 that constructing classifiers that are robust to specific distributions is a means of overcoming the ostensible barrier to constructing robust classifiers imposed by our results. However, we believe our results still have important consequences: for example, there is no guarantee that malicious agents interacting with computer vision systems will restrict themselves to some natural image manifold.
>
> We also point out that we can adapt our bound to account for image distributions as discussed in Section 3.2, though at the cost of a multiplicative factor that is difficult to quantify.
>
> *“it's unclear how applicable such results as these are to building the intuition of robustness researchers at large, given that it's generally believed that images live on a lower-dimensional manifold (and therefore have measure 0 in ambient space).”*
>
> As this and other reviewers have pointed out, this work is mainly theoretical, and hence its value lies in providing an important bound in robustness theory. However, we do believe that the form of the bounds could potentially lead to insights, such as how the bound of $O(n)$ is suggestive of the saliency of line drawings.
>
> We additionally note that since the image spaces we deal with are finite, the techniques we employ are able to circumvent the problem where the image manifold is of measure zero. This may provide a means of better understanding the distribution of natural images in future work.

---

### Official Review · Reviewer_vaeu · 2022-10-24

**Confidence:** 3
**Correctness:** 3
**Technical Novelty And Significance:** 2
**Empirical Novelty And Significance:** Not applicable
**Recommendation:** 5

**Clarity, Quality, Novelty And Reproducibility:**

The paper is generally well written and easy to follow.
Regarding the notation, it isn't clear why inputs are parametrized by a height `n` and an aspect ratio `q` (for images of size n x (nq)), instead of simply considering images of size n x m.

It also isn't clear why the paper focuses on *image* classifiers, when all the results are presented in an generic way that is independent of any specific properties of images.
E.g., wouldn't the same results apply to audio classifiers, or time-series classifiers, or video classifiers, etc?

The role of the "junk class" (if present) could be clarified. As far as I understand, the robustness results hold for any non-junk class. But is there any distinction between perturbations that send an image from a (non-junk) class C1 to the junk class, versus a perturbation that sends an image from a (non-junk)  class C1 to a different (non-junk) class C2? These two types of "adversarial examples" seem qualitatively different.

**Strength And Weaknesses:**

Strengths:
- Interesting and tight result on the robustness of arbitrary classifiers
- The paper is easy to read

Weaknesses:
- Not clear why the focus is on images, as the result seems to apply to arbitrary classification tasks
- Unclear what the takeaway of the result should be. The paper seems to suggest that the result is some kind of fundamental limitation of robust classification, but Figure 2 simply seems to show that Lp norms are poor measures of semantic similarity (and thus poor measures of robustness as well). This is not a new insight.


**Summary Of The Paper:**

The paper studies fundamental limits on the Lp robustness of a classifier.
The main result is that under natural assumptions, any classifier on n x n images will be vulnerable to L2 perturbations of size O(sqrt(n)).
The paper further shows that the result depends on the chosen bit-depth of images.

**Summary Of The Review:**

The main result of the paper seems technically sound, and the approach based on Hamming graphs seems novel and interesting.
The actual interpretation of the result is less clear to me.

When comparing the robustness bounds to "typical distances between random elements of the image space" in Section 3.1., are these distances between images from the junk class? If so, this may not be the most interesting measure to compare against. The typical distance between elements of "interesting" classes would be the more appropriate point of comparison here.

Sections 3.2 and 3.3 try to give some interpretations of the result and implications for robust vision systems, but ultimately it seems that the right conclusion here is simply that drawn at the end of section 3.2: "Another way of circumventing the barrier to constructing reliable computer vision systems imposed by our bounds is to note that a perturbation with a small p-norm is not necessarily imperceptible"
Examples such as the ones in figure 2 are not novel: similar argument were made in an early paper by Fawzi, Fawzi and Frossard which should definitely be considered in the related work: https://arxiv.org/abs/1502.02590
It has been known for a while that Lp norms are not good a metric for perceptual similarity. And so we know that small Lp perturbations can at times change classes, while large Lp perturbations might not (see e.g., https://arxiv.org/abs/2002.04599)

In light of this, it is not particularly clear how the paper's main result should be interpreted.
Ultimately, it is is simply a result about how input spaces can be partitioned to maximize distance between classes. Whether this result has any implications for robust classification is unclear.

---

> ### Author Response · Authors · 2022-11-15
> **Response to reviewer**
>
> We thank the reviewer for their thoughtful comments. We would like to respond to the following comments and questions from the reviewer, some of which we paraphrase due to space limitations:
>
> *“In light of this, it is not particularly clear how the paper's main result should be interpreted. Ultimately, it is simply a result about how input spaces can be partitioned to maximize distance between classes. Whether this result has any implications for robust classification is unclear.”*
>
> The main result of the paper is a theoretical one, namely that the geometry of image spaces imposes fundamental limits on the classifiers one can construct over them. As noted by a separate reviewer, it does not necessarily say that robust classification is impossible from a practical standpoint, but rather it provides a simple but important lower bound to robustness theory which can inform how we formulate practical notions of robustness.
>
> We explore means of circumventing these fundamental limits in Section 3.2, either by focusing on specific image distributions or by constructing better notions of distance. In the case of the latter, we should note that Tramèr et al. (https://arxiv.org/pdf/2002.04599.pdf) have pointed out that finding a distance that aligns with semantic saliency essentially solves computer vision. In the absence of such a solution we believe there is great value in understanding the properties of surrogate distances like those based on Lp norms.
>
> *“The paper seems to suggest that the result is some kind of fundamental limitation of robust classification, but Figure 2 simply seems to show that Lp norms are poor measures of semantic similarity (and thus poor measures of robustness as well). This is not a new insight.”*
>
> We would like to clarify that our examples in Figure 2 are just examples of brittle features that are salient, although they are interesting in providing a possible interpretation to our theoretical results.
>
> The actual existence of such features is based on our bounds as discussed in Section 3.3. The inadequacies of Lp norms have indeed been noted in prior work, but to our knowledge ours is the first to base this observation on theoretical bounds. This theoretical foundation allows us to quantify these inadequacies precisely, and has the potential to drive further insight since not all consequences are immediately obvious.
>
> *“When comparing the robustness bounds to "typical distances between random elements of the image space" in Section 3.1., are these distances between images from the junk class? If so, this may not be the most interesting measure to compare against. The typical distance between elements of “interesting” classes would be the more appropriate point of comparison here.”*
>
> The typical distances discussed in Section 3.1 are measured between images from the entire image space regardless of class. We have revised the paper to include an empirical measurement of typical distances between images from a subset of Imagenet, which can be thought of as a surrogate for elements of “interesting” classes. These measurements yielded similar values, and the added results can be found in Appendix A.5.1.
>
> *“Why are the images parameterized by height $n$ and aspect ratio $q$?”*
>
> We use $n$ and $q$ to emphasize the idea that the width and height should be roughly similar in magnitude, which holds as long as $q$ is not too large or too small. We have added a footnote at the start of Section 2.1 to note this.
>
> *“The results are independent of specific properties of images.”*
>
> Our theoretical results are indeed slightly more general than image spaces and apply to data belonging to any space that can be expressed as a cartesian product of equally sized bounded intervals. We revised the paper to emphasize that this is the case at the start of Section 2.1. We note that this condition is not met by arbitrary data, so we have opted to keep the main discussion on image classification due to its importance.
>
> *“What is the role of the “junk class”, and is there any distinction between perturbations that move images to junk classes and those that move images between interesting classes?”*
>
> Image classes that are not interesting image classes (junk classes) can be made robust - for example they can potentially cover the entire image space. Our purpose in defining them is to eliminate these pathological cases. We have amended Section 2.1.1. to clarify this. Our theoretical results only consider the source of the image and not the destination, so they don’t have much to say about whether the destination of the image is interesting or uninteresting.
>
> Finally, we thank the reviewer for their suggested additions to our literature review, and have revised Section 1.1 to include these additions. We hope that given these revisions and clarifications the reviewer would consider raising their score.

---

### Official Review · Reviewer_K2cJ · 2022-10-24

**Confidence:** 4
**Clarity, Quality, Novelty And Reproducibility:** The paper is well written, novel and …
**Correctness:** 4
**Technical Novelty And Significance:** 4
**Empirical Novelty And Significance:** Not applicable
**Recommendation:** 8

**Strength And Weaknesses:**

# Strengths :
1) The analysis appears to be technically sound and best one could seek to establish under this setup.
2) The incorporation of discrete image space via expansion properties of hamming graphs was insightful and could spurn other results.

# Weakness :
The authors favorably view their work in comparison to cited work Fawzi et.al. 2018. I agree that the result here is better for its extension to any entry-wise p-norm and restriction to discrete image spaces. However I contend that the distribution-agnostic flavor of the result is a weakness rather than a strength. For eg.
1) there is an implicit assumption that the union of pre-images of class labels C^{-1}(y) should cover the entire discrete image space.
2) Real data distributions are certainly not supported on any possible image in I_{all} and requiring a uniform level of robustness at any such image is a pessimistic objective to begin with.

There are implications from these points that weaken the final inference of the stated results.

- The statement "w.h.p of uniformly an image from C^{-1}(y)" now includes images that humans would ostensibly consider random noise but that which a classifier might faultily predict label "y". In fact every random noise or approximately random noise image is present in the set I_all. The authors indeed indicate this -

_"Our bounds do not immediately preclude the existence of small fractions of images within interesting image classes that are robust to large perturbations, and it is possible that those are precisely the set of images that are commonly encountered in deployment. Therefore, our bounds do not directly prevent the construction of classifiers that are robust with respect to some given image distribution"_

- Ideally the notion of *interesting classes* should capture the data-distribution as subsets of C^{-1}(y) that exclude regions of input that have measure zero. The current characterization of interesting classes is weak in this regard.

- The expected distance between images from real data could be far smaller than the expected distance between any two uniformly sampled images in I_{all}. This could render the adversarial energy at which a point is non-robust much larger relatively. For example the denominator in LHS of equation (2) might be much smaller and hence the upper bound might not vanish or vanish more slowly.

- Further the quality of such a result in a non-asymptotic setting is more relevant as real image data typically have n <= 224 (outside of medical imaging). When n -> infinity, I might expect a standard concentration phenomena that places more weight on exterior /edge of I_{all} which further worsens the lack of distributional information. My intuition is that the gap between expected distance between randomly sampled data under a specific data distribution P vs expected distance between uniformly sampled images grows with n making the analysis potentially looser asymptotically in the context of real data. (Note : This last thread of logic is unverified informal intuition and I would be happy to see arguments for why it is not the case).

- To summarize, here's a statement from the paper -

_"Then for most images, a tiny perturbation can make the given image trigger undesired behaviour, ostensibly making the classifier unreliable."_

It is my opinion I think the result does not make a statement for *most* images of a real data distribution and the adversarial energy might not be *tiny* in comparison to expected distance between images of different classes.

As a *concrete actionable feedback* I suggest the following experiment - evaluate the expected distances between randomly sampled train/test data from a benchmark data set (say CIFAR10) and compare it with the theoretical limit under uniform sampling from I_{all}.
Further one could observe this distance as images are scaled to larger spatial dimensions (which act as proxy for a true image distribution with larger n). These experiments could bolster the effectiveness of the main result.


- As a final observation, Section 3.3 aims to demonstrate that brittle features (such as line-drawings) can impart non-trivial semantically salient information. I agree with this observation but would like to point out an asymmetry : real data aren't well approximated as random noise + line-drawings. Image (c) in Figure 2 shows that removing (a) can fundamentally change the semantic content but Image (e) shows that this isn't necessarily the case for real images. So, removal of brittle features might not remove all semantically salient information even if addition can impart them. I contend that an ideal characterization of meaningful images should exclude Image (c) as the signal-to-noise ratio is low.



**Summary Of The Paper:**

This work seeks to understand the limits of robustness of any image classifier. The framework here considers images as discrete objects on the grid space of (spatial dims n  x channels h ) with each entry in [0,1] representable by a finite bit string of "depth" b.
Let I_all denotes all possible images that can be represented as described. Each classifier induces a partition of I_all based on predicted labels.

The authors make the following novel observations that are independent of the image distribution and any properties of the specific classifier (beyond the induced input partition) -
1) Most images in any class partition can be shifted to an image in a different class partition by a perturbation that is O(n^{1/max{p,1}). Or w.p. at least (1-delta), any uniformly sampled image in this class is vulnerable to perturbations of size O((\sqrt{n^2 h log(2/delta) })^{1/p} ) for p in [1,\infty).
The required energy of perturbations vanish in comparison to the expected distance between any two uniformly sampled image from I_all.

2) The above statements are asymptotically optimal as image size n -> infinity. Thus better rates require further assumption on the data distribution or the classifier properties.



**Summary Of The Review:**

This work shows a fundamental limit of robustness for image classifiers, albeit in a worst-case setting where all possible discrete images of a certain size are considered. The main result is asymptotically optimal and leverages novel techniques using expansion properties of Hamming Graphs. While rate optimal within the framework considered, the results demonstrate that understanding robustness requires further accounting for properties of data distribution or indications of such via the trained classifier.

---

> ### Author Response · Authors · 2022-11-15
> **Response to reviewer**
>
> We thank the reviewer for their thoughtful and insightful comments, such as those exploring more fine-grained notions of interesting classes and semantic saliency, and for their favorable review. We would like to respond to the following comments:
>
> *“The authors favorably view their work in comparison to cited work Fawzi et.al. 2018. I agree that the result here is better for its extension to any entry-wise p-norm and restriction to discrete image spaces. However I contend that the distribution-agnostic flavor of the result is a weakness rather than a strength.”*
>
> We would like to clarify that the major strength in having a distribution agnostic result is that our bounds are unambiguous and can be computed exactly. We agree with the reviewer that there are merits to bounds that account for image distribution, however parametrization on image distributions often leads to uncertainties and ambiguities due to their complexity and our incomplete understanding of them (except in toy cases). Nevertheless, in Section 3.2 we do discuss how we can employ the techniques of Fawzi et al. to obtain a distribution specific bound at the cost of having a dependency on a “modulus of continuity” as they do. As discussed, this dependency allows the construction of classifiers that are robust against certain image distributions, such as the toy example we describe in that section.
>
> *“As a concrete actionable feedback I suggest the following experiment - evaluate the expected distances between randomly sampled train/test data from a benchmark data set (say CIFAR10) and compare it with the theoretical limit under uniform sampling from I_{all}.”*
>
> We thank the reviewer for the actionable feedback, and have computed the average distances within Imagenette (a subset of Imagenet) per their advice. Interestingly, it turns out that for the dataset in question the average distances remain similar to the average distances within the entire image space, though they are slightly lower. We have added this observation and the one above to the main text in Section 3.1, and have included the results in Appendix A.5.1 along with a detailed explanation of the computation.
>
> *“Further one could observe this distance as images are scaled to larger spatial dimensions (which act as proxy for a true image distribution with larger n). These experiments could bolster the effectiveness of the main result.”*
>
> We scaled the Imagenette dataset to various sizes, and interestingly there also appears to be little noticeable difference in typical distances when we vary the sizes of the images. This additional observation is also included in Appendix A.5.1.

---

### Official Review · Reviewer_mJYy · 2022-10-31

**Confidence:** 4
**Correctness:** 4
**Technical Novelty And Significance:** 3
**Empirical Novelty And Significance:** Not applicable
**Recommendation:** 8

**Clarity, Quality, Novelty And Reproducibility:**

### Clarity
* Overall the paper is well written. The exception is the usage of $n^2qh$ to denote the dimension instead of e.g., $d$ or $n$. The current notation is just annoying and I see no benefit of using it. I highly advice the authors to consider simplifying notation here. In a similar spirit, I think Algorithm 1 could be replaced with $f(x) = \text{sign}(\mathbf{1}^T x - d/2)$ for image $x$ for better readability.

* Consider providing a "dual" table to Table 1, where we fix the perturbation budget and ask what fraction of an interesting class can be robust now. Specially it would be interesting to provide such results for images of CIFAR/ImageNet dimensionality and the standard radii. E.g., $\ell_\infty \sim 8/255$,  $\ell_2 \sim 1$ and $\ell_1 \sim 2$.


### Novelty
* The proof technique adapted from previous works  seems to be novel in this field. The crucial tool of the paper is Theorem 3 of (Harper 1999). Unfortunately, the referenced paper states "We leave the verification of this as an exercise for the reader." The reader could not verify it by themselves.


* While I think the paper is novel enough, the relevant literature is cited, but not sufficiently credited. E.g., this paper claims superiority over the previous approaches since the previous work produced results dependent on the data distribution, while here the result is independent of the image distribution. On the other hand, the previous approaches  derived results for uniform distribution, which is equivalent to the setting assumed here. Similarly, the result for hypercube is mentioned in the literature, but the provided literature review here suggests that considering discrete input is a new concept. The literature also already showed that the interesting things happen when the perturbation is of the size $\sqrt{d}$; thus, the results of this paper are somewhat unsurprising. That being said, I think the article addresses an interesting problem and as far as I can see, it doesn't follow directly from the literature. However, the literature could be credited more fairly. The results I was referring to can be found in the $3$ papers discussed in the last $2$ paragraph of sec 1.


### Quality
* There are minor problems listed below, but overall the results look correct (did not check thm 3 yet) and the problem solved is interesting. On the other hand, I do not see any direct applications of the results - this is somewhat expected for a theoretical work.

### misc
* Thm 1 the quantifier do not look right to me. The result holds when we consider any $c$, not just when we consider all of them.
* Thm 2 consider stating $1/4 < c < 0$.
* eq. (10), there should be $p^i(1-p)^{n-i}$, but there is $k$ instead of $i$.
* just before eq. (13). "by induction we have" - we don't really have it, it is the induction hypothesis.
* eq. (15), in denominator sould be $(1-p)^{n-x-1}$, later the same error in eq. 17 (also see typo in eq. 17, $U_n$)
* In lemma 4 we start with a given $r$ but later we define $r$ to be a median of the distribution and in the end we reference the original $r$ again. Please, make it unambiguous. Further, we needed to assume that $np+1-k  < E(X) = np $ so that we use Hoeffding's inequality; that is, $k  > 1$.
* Eq (21): there should be $\leq$ instead of $=$.
* Thm 4. There should be "path with at most X edges", not just "path with X edges".
* For eq. (25), my calculations yielded $(1/2)e^{\dots}$, while there is $2e^{\dots}$. Please check.
* eq. (26) is not correct, there should be (e.g.,) $(c+1)^2$ in the exponent.
Note that these two errors are propagated in many places in the paper.
* the very last paragraph before sec A.1.3: "But then $... > 2e^{-2c^2}...$ there should be no $-$ in the exponent.
* Lemma 5 - the statement is missing $q$ in the perturbation size. The proof is a simple corollary of the previous theorem and In my eyes could be reduced to one short paragraph.
* Why do we often have cases $p \geq 2$? I think it holds when $p \geq 1$; admittedly, I did not pay much attention on the case $1 \leq p \leq 2$ when checking the proofs.
* $p$-norm is norm only when $p\geq 1$. However, you use it for any $p$. Please, mention this fact when introducing $p$-norms.

(Harper 1999) On an isoperimetric problem for Hamming graphs



**Strength And Weaknesses:**

The paper considers image space to be quantized. Then they form a graph from it, in a way that every possible image is a vertex and two vertices are connected iff they differ at a precisely one position. Such graph is called a Hamming graph and the distance of two vertices correspond to $L_0$  "distance" of the corresponding images. Thus, the original problem of - how big the adversarial budget have to be in order to make the majority point not robust - becomes similar to a so-called isoperimetric problem on Hamming graphs, and finally the result is obtained for adversary with $L_0$ budget. Since the considered domain is bounded, one can simply compute the minimal radius of $L_p$ ball containing the $L_0$ "ball" of given radius.  (thm 1)

Later, a construction of a classifier is given for which the $L_1$ adversary can find just a few points that are not robust given certain budget, where the property is proved using anti-concentration inequality. Finally, the result is extended to other $p$-norms by calculating the budget of $L_p$ adversary that is contained in the given $L_1$ budget. (thm 2)

Finally, the role of quantization levels is investigated yielding an alternative upper bound for the robustness when $p \geq 2$.  (thm 3)

### good points
* The considered problem is quite natural and interesting.
* The techniques used here seem to be novel in the field.
* The paper provides both lower-bounds and upper-bounds for possible robustness and they are quite matching.
* Clear exposition of proofs.

### ungood points
I believe they all can be fixed and later I provide more details on them.
* The literature is not fairly credited.
* Unnecessarily complex notation. An image could be a $d$-dimensional vector, now it is tensor of order $4$.Thus there are used unnecessary letters and they are sometimes overloaded ($n,q$ in the proofs have different meanings than they have in the main paper.
* The result is disconnected from practice by considering by orders of magnitude larger perturbations than is commonly used.




**Summary Of The Paper:**

The paper studies fundamental limits on robust classification in quantized domain $\mathcal{D \subset [0,1]^n}$. To this end, it is shown that whenever a classifier $f$ classifies less than half of all the points as class $c$, then almost no input $x$ such that $f(x) = c$ is robust in the sense that there exists an adversarial example $x'$ such that $||x-x'||_p \leq O(\sqrt{n})$ and $f(x') \neq c$. It is also often shown that $O(\sqrt{n})$ is asymptotically optimal. These results are presented for $p \in \set{0, 1, \geq 2}$.

**Summary Of The Review:**

The paper shows that $O(n^{1/{2p}})$ is the maximal perturbation budget for which a reasonably robust classifier may exist for inputs from a subset of $[0,1]^n$. The techniques used are new in this field and overall I think this is an interesting problem with reasonably complete solution - there is still one case that is not fully answered - upper-bound for $p \geq 2$.



If the errors in derivations are fixed and the statements are corrected (see misc of previous section), I will raise my score. I will also additionally raise the score if my other concerns are addressed/discussed satisfactorily (see ungood points and their details in the previous section - specially the related work part).

---

> ### Author Response · Authors · 2022-11-15
> **Response to reviewer (ungood points)**
>
> We thank the reviewer for their thoughtful comments. We apologize for the various issues identified by the reviewer, and greatly appreciate their extraordinary thoroughness in identifying them. We have fixed them in the revision, and will enumerate the fixes in a separate reply due to the character limitations.
>
> In this reply we would like to respond to the ungood points raised by the reviewer:
>
> Regarding the reviewer’s concern that the literature is not fairly credited: we did not intend to suggest that considering discrete input was a new concept - our intended statement was that we showed how discretization fundamentally changes the robustness properties of the image space. We believe that this is a new insight. To avoid confusion, we amended the paragraph discussing the work of Diochnos et al. to emphasize that they investigate discrete inputs and that their results are not parametrized by specific data distributions. We have also added a footnote to the previous paragraph to further deter misconceptions. We have further amended our literature review to state the bound that Diochnos et al. derives is proportional to the square root of the data dimension, although we note that for $p \geq 2$ our bounds are smaller than the root of the data dimension. We hope this revision helps clarify the state of the existing literature and more accurately contextualizes our contribution.
>
> Regarding the reviewer’s concern about the complexity arising from representing images as tensors and about notation in general: while representing images as tensors adds some complexity, we believe there is value in having the theorems stated in this form in the main body of the paper. This way the general reader can immediately understand the relation between side lengths, number of channels, and bit depth to the bounds, and can immediately apply them without any additional work. However, we understand that the notation can be cumbersome especially in the proofs. We have endeavored to keep out the various parameters associated with tensors until the end of the proofs, where the exact forms of the theorems are derived. The exception previously was Theorem 3, which we have now revised such that the data dimension $N$ is used instead of $n^2qh$. For additional clarity, we have revised Appendix A.1.2 to use $w$ instead of $q$, we have revised the main body of the paper such that $n^2qh$ and $\sqrt{qh}*n$ appear consistently throughout, and we now note that the quantity often appears together and can be thought of as the data dimension at the start of Section 2.1. We have opted to leave Algorithm 1 as is for additional clarity for the general reader, but we appreciate the reviewer’s suggested revision and have adapted it in a footnote where the algorithm is introduced for readers familiar with the notation.
>
> Regarding the reviewer’s suggestion for providing a ``dual’’ to Table 1 and their observation that our bounds consider perturbations that are larger than ones considered in practice: we have added a figure that plots the relations in Table 1 in Appendix A.6 for various fixed parameters of interest, which provides a means of inverting those relations as suggested by the reviewer. Our bounds do not say interesting things about distances on the order of 1 or 2 since our proofs often loosen the bounds by such amounts to deal with certain details like rounding. We imagine our bounds can be made tighter by constant factors or additive constants; however we note that it is not too unexpected for classifiers to have much lower robustness in practice - the bounds we derive are still exceptionally small when one considers it is the highest achievable for arbitrary classifiers.
>
> We hope that given these revisions and clarifications the reviewer would consider raising their score, and will be happy to address any further points raised by the reviewer.

---

> > ### Author Response · Authors · 2022-11-15
> > **Response to reviewer (misc)**
> >
> > We have enumerated our fixes to the issues pointed out by the reviewer in the "misc" section below, in the order they were raised by the reviewer:
> >
> > - Theorem 1 has been reworded to make it clear that $c$ is arbitrarily fixed first.
> >
> > - A remark has been added to note that Theorem 2 is trivial if $c \geq 1/4$.
> >
> > - Equation (10) has been corrected to say $i$ instead of $k$.
> >
> > - The wording before Equation (13) has been revised to “We can prove the other cases by induction on $x$” - the purpose of this sentence is to indicate that $x$ is the variable induction is carried out on.
> >
> > - The denominators in Equations (15) and (17) have been fixed.
> >
> > - In Lemma 4, $(k-1)$ has been changed to $\max(k-1,0)$ to ensure the relation holds. This has also been updated in its subsequent application in the proof of Theorem 4. We have also changed $r$ to $m$ when discussing the median to avoid ambiguity.
> >
> > - We have fixed equation (21) to properly display $\leq$ instead of $=$.
> >
> > - The statements in Theorem 4 all appear to be of the form “path with X edges or less”. Please let us know if we are mistaken.
> >
> > - Regarding Theorem 4 and Equations (25) and (26) and the last line before the start of Section A.1.3: Theorem 4 is correct, but as the reviewer points out there were some issues in the analysis. This has now been fixed. The crux of the issue was that the application of Lemma 4 involved some equation rearrangement that was performed incorrectly. When done correctly, Equation (26) does follow, since $e^{(c+1)^2} > e^{c^2}$ as long as $c \geq 0$. The issue with the rearrangement was canceled out by the complementary error right before the start of Section A.1.3, which we have also fixed, yielding the original theorem statement. Since the original theorem statement is correct, the error should be contained to the proof and should not have propagated throughout the paper. We remark that Theorem 4 is a fairly intuitive consequence of Lemmas 3 and 4, hence its correctness despite the noted issues.
> >
> > - We have fixed the statement of Lemma 5 to include $q$. We have prepended a short paragraph to the proof to indicate and explain that it is a simple corollary of Theorem 4. We have opted to keep the remainder of the proof for readers who may prefer additional details spelled out.
> >
> > - The case of $p \geq 2$ is interesting because this is where the bit-depth dependencies kick in. When we take the limit of the bit depth to infinity, the required perturbation budget to switch classes shrinks to a constant independent of data dimension. This does not hold for the case of $p \leq 1$ as demonstrated by Theorem 2, so we separate these cases. As noted in Section 2.1.2, we only consider non-negative integer values of $p$, although much of our analysis does carry over to the non-integer case. We believe it should be possible to show that for the classifier constructed for Theorem 2, the required perturbation budget to switch classes does not shrink to a constant independent of data dimension for any $p < 2$ including non-integer values.
> >
> > - We have amended Section 2.1.2 to emphasize that the 0-"norm" is not a norm.
> >
> > We once again thank the reviewer for their efforts, and hope that they will consider raising their score in response to these revisions. We would be happy to address any further comments as well.

---

> > > ### Comment · Reviewer_mJYy · 2022-11-16
> > > **response**
> > >
> > > Thanks for the comments, It looks like the "typos" are fixed.
> > > * thm 4 looks good, I probably overlooked it.
> > > * the second to last point - Maybe I don't remember the proofs correctly, but my point is that you don't say anything about the case $1 < p < 2$ - while I thought that your analysis for the case $p \geq 2$ applies here. Maybe it does not and I just don't remember it.
> > >
> > > Regarding the literature; I think it is much better to write down all the nice results that were discovered before (since they are really relevant) and then tell what are the new things. I appreciate the effort of the authors to improve the section, but I would still like them to consider to expand the section and show concrete results that were obtained in the literature.
> > >
> > > Regarding the notation: I still don't like it but the notation is your choice. Specially "This way the general reader can immediately understand the relation between side lengths" is true, but I believe it would be true even when you write just dimension $d$ :-) E.g., look thm 1. It is impossible to guess the meaning of the letters correctly. On the other hand, if there would be just $d$, then the main theorems can be understood without much of pain. Also using $*$ to denote (scalar) multiplication is not my favorite choice.
> > >
> > > Regarding the "dual" bounds: Thanks, I think they should be in the main paper, because I consider this to be the question one expects to be answered (in how large robust radii can we hope?)  I also think it could be good to reference it in a more "expected" place, such as caption of tb 1. (in table 1 $ln$ -> $\ln$, maybe also do bigger brackets there.)
> > >
> > > It would also be great if we could have the proof of (Harper 1999) but I don't know if it is feasible.
> > >
> > > Overall, I'm fairly satisfied by the answers and I'm raising the score, although I hope there will be additional changes in the final version.

---

> > > > ### Author Response · Authors · 2022-11-19
> > > > **Further response**
> > > >
> > > > We thank the reviewer for their thoughtful and helpful comments in improving the paper, and for raising their score. We have taken steps to address the additional comments in the new revision we have just posted, and will take them into account for any further revisions. Specifically:
> > > >
> > > > Regarding the literature: We have made additional efforts to more clearly exposit past results. Specifically, we now elaborate on the relevant contributions of the other work we touch on at the start of Section 1.1 in a footnote. Furthermore we now note that Fawzi et al. numerically evaluate their bounds, we’ve revised the text to specify that Diochnos et al. showed that the probability of misclassification can be made arbitrarily high, and we now specify that the work of Mahloujifar et al. generalizes the result of Diochnos et al. to Levy families, and that their bound is of the exact same nature as that of Diochnos et al. We believe together these changes make clear what the previous results are before describing how our results differ. We would be happy to address further suggestions, although we do note that we are now quite close to the page limit.
> > > >
> > > > Regarding our notation: We thank the reviewer for pointing out that the meaning of the symbols may be hard to guess, and have augmented the theorem statements so that they clarify the meanings of the symbols in a self contained way to further facilitate use by the general reader. We appreciate the reviewer’s point of view and understand the merits of their suggested notation, but still feel that there is value in the current notation.
> > > >
> > > > Regarding the "dual" bounds: We have added another reference to the new figures in the caption of Table 1 as suggested by the reviewer. However, due to space limitations we are opting to keep the figures in the appendix, as they are quite large.
> > > >
> > > > Regarding some other specific comments:
> > > >
> > > > *“It would also be great if we could have the proof of (Harper 1999) but I don't know if it is feasible.”*
> > > >
> > > > We understand and appreciate the reviewer’s concern about Harper seemingly leaving the proof of Theorem 3 as an exercise. We believe Harper’s intention could be better reworded as “the proof of Theorem 3 is similar to the proof of the main result”. It seems like a full proof of Theorem 3 would essentially be a reproduction of Harper’s paper, so we feel it would only be appropriate to include such a proof if we were able to significantly simplify it, which at this point we have not achieved.
> > > >
> > > > *“the second to last point - Maybe I don't remember the proofs correctly, but my point is that you don't say anything about the case  - while I thought that your analysis for the case  applies here. Maybe it does not and I just don't remember it.”*
> > > >
> > > > Yes, we do not say anything about the case where $1 < p < 2$ since we do not consider non-integer values of $p$. However the reviewer is correct in observing that our techniques for generalizing $p$ hold for non-integer values of $p$ as well (Lemma 6, Lemma 11, and Lemma 15), so our proofs could be extended to discuss the non-integer case. Our previous reply contained some speculation on what might be the case if an additional result investigating the case of $1 < p < 2$ were done.

---

### Decision · Program_Chairs · 2023-01-20

**Decision:**

Accept: poster

**Justification For Why Not Higher Score:**

Interesting theoretical result but the practical value is limited

**Justification For Why Not Lower Score:**

The result is an interesting contribution on upper bounds on robustness which are shown to be tight.

**Metareview: Summary, Strengths And Weaknesses:**

The authors provide upper bounds on the robustness of an image classifier (images are considered as quantized) and show that there exists a classifier for which this is basically tight. The advantage compared to prior work on this topic by Fawzi et al and Diochnos et al is that there are no distributional assumptions (which can also be seen as a weakness) and that there are results for all $l_p$-balls (including $p=0$).

Strengths:
- novel result on robustness which is shown to be tight
- use of novel proof techniques
- clearly written

Weakness:
- the notion of interesting class as the pre-image set of a classifier is questionable, in particular as the support of true image classes is likely to be concentrated around a low-dimensional structure
- interpretation/discussion of the results could be further improved (even though the authors have improved this during the rebuttal)

All reviewers except one argue for acceptance. I think that this is a nice theoretical result which adds to our understanding of adversarial robustness, in particular also regarding discretization. The practical implications are unclear and the authors are honest about them.

I strongly recommend to the authors to take the comments of the reviewers on the notation and presentation into account. I don't see any reason to keep this notation, in particular as one reviewer pointed out that there is nothing special about images in the construction of the proof. Also the authors should work to make their main result more accessible, in particular the direct connection to Theorem 3 in Harper, 1999 and their "adaptation" should be made clear, that is by citing the original result as it is and then extending it to the stated results using their notation (currently the correspondence is not obvious). Even better would be to give the complete proof, as Theorem 3 is an exercise for the reader in the paper of Harper.

In the discussion of Section 3.3 the references provided by the reviewers have to be added which discussed similar aspects.

**Note From Pc:**

if the above contains the word "oral" or "spotlight" please see: "oral" presentation means -> notable-top-5% and "spotlight" means -> notable-top-25%. As stated in our emails, we are disassociating presentation type from AC recommendations